# The Geometry of Sequential Learning:
# Lie-Bracket Prediction of Transfer Order

**John Sweeney** [1]

## Abstract

Sequential learning is order-dependent: from Pile-style next-token domain adaptation to instruction-SFT and DPO, $N$ candidate sources induce $N!$ possible curricula. We show that the local order effect is governed by a computable geometric quantity, the Lie-bracket commutator of gradient update fields, yielding a pairwise score for whether $A \to B$ or $B \to A$ is better for a target domain. The pairwise bracket primitive also defines a *Lie-Bracket Tournament*: with a shared $\theta_0$ target-gradient reference, Hessian symmetry gives Borda/row-sum scores from one Hessian-vector product per source, $O(N)$ dot products, and an $O(N \log N)$ sort, without materializing the $O(N^2)$ edge matrix. Empirically, the planner reaches 98.1%/98.9% pairwise accuracy at $k=1$ for instruction-SFT/DPO, remains at 73.1%/72.2% at $k=20$, and preserves the original pretraining-domain evidence with 82.4–92.0% accuracy across four LLMs and 91.1% on diffusion. At curriculum scale, it recovers the best of all $3!$ schedules in 87.5% of trials, ranks 85 Stack programming-language source domains for a Python target in the 99th sampled percentile, and reaches the 99.0–99.6th sampled percentile on 56 MMLU subjects, sharply above the reported descending gradient-norm baseline. These results reframe sequential learning as a geometric tournament problem: commutators provide both local pairwise order information and a scalable primitive for many-domain schedules.

## 1. Introduction

Sequential post-training pipelines rarely adapt a model on a single homogeneous dataset. A model may be instruction-tuned on several task families, preference-tuned on several feedback domains, and finally adapted or evaluated on a target distribution. In such pipelines, the order of datasets is not a cosmetic implementation choice: training first on one domain and then another can change downstream target loss. For two source domains $A$ and $B$, one can still run both $A \to B$ and $B \to A$. For $N$ sources, however, exhaustive curriculum search requires evaluating $N!$ orders. This is the central practical obstacle.

The central observation is that order dependence is structured. Gradient-based training defines nonlinear update operators, and nonlinear operators generally do not commute. When the update operators for two datasets fail to commute, composing them in opposite orders traces different trajectories in parameter space and can change target loss. This is the same geometric phenomenon that appears in operator splitting and numerical dynamics, where the leading-order error is governed by a commutator, or Lie bracket (Hairer et al., 2006). We use this geometric lens to turn local order effects into a practical planner for sequential learning.

**Pairwise geometric primitive.** Let $L_D(\theta)$ be the loss on domain $D$, with gradient $g_D(\theta) = \nabla L_D(\theta)$ and Hessian $H_D(\theta) = \nabla^2 L_D(\theta)$. A single gradient step on $D$ is the update operator $U_D(\theta) = \theta - \eta g_D(\theta)$. For two sources $A, B$, the two compositions $\theta_{AB} = U_B(U_A(\theta_0))$ and $\theta_{BA} = U_A(U_B(\theta_0))$ satisfy

$$\theta_{AB} - \theta_{BA} = \eta^2 \big( H_B g_A - H_A g_B \big) + O(\eta^3), \quad (1)$$

with all quantities evaluated at $\theta_0$. Thus the bracket vector $b_{AB} := H_B g_A - H_A g_B$ is the leading driver of order dependence. Projecting this displacement onto the target gradient gives the directional score

$$\sigma_{AB}^{(E)} := \langle g_E, b_{AB} \rangle, \quad (2)$$

whose sign predicts whether $A \to B$ or $B \to A$ gives lower loss on target $E$. We reserve $\omega_{AB}^{(E)}$ for the normalized cosine version of this alignment, so $|\hat{\omega}|$ is a scale-free confidence score while $\eta^2 |\hat{\sigma}|$ estimates stakes.

**Drift-matched scoring.** The target gradient changes along the shared first-order drift induced by $A$ and $B$. Rather

[1]Sideplane AI. Correspondence to: John Sweeney <john.sweeney@sideplane.ai>.

*Proceedings of the $43^{rd}$ International Conference on Machine Learning*, Seoul, South Korea. PMLR 306, 2026. Copyright 2026 by the author(s).

than evaluating $g_E$ only at $\theta_0$, our default estimator evaluates it at the Trotter reference point $\theta_{\mathrm{ref}} = \theta_0 - \eta(g_A + g_B)$. This point approximates the common drift of both orders without running either complete order, reducing target-gradient drift error while preserving deployability.

**From pairs to Lie-Bracket Tournaments.** The pairwise score becomes a many-domain scheduler once each pair of source domains is treated as a tournament edge. For a fixed target $E$, the score for $D_i$ versus $D_j$ says which one should go earlier and by how much. The collection of pairwise preferences forms a weighted *Lie-Bracket Tournament*; ranking the tournament turns the original $N!$ curriculum search problem into a ranking problem over $N$ domains. The computational simplification is substantial: in the scalable tournament variant we use a shared $\theta_0$ target-gradient reference, so Hessian symmetry gives all Borda/row-sum scores from the source gradients $g_i$ and one Hessian-vector product $u_i = H_i g_E(\theta_0)$ per source. The full pairwise matrix need not be materialized for Borda/row-sum ranking, so score computation is linear in $N$, followed by an $O(N \log N)$ sort. This shared-reference choice is different from the drift-matched pairwise planner; it is what makes tournament edges $N$-independent and exact row-sum aggregation cheap. Exact permutation scoring remains available for small $N$.

**Evidence and scope.** The experiments are organized around the same progression. First, instruction-SFT and DPO show that the bracket transfers beyond the original next-token setting: pairwise sign accuracy reaches 98.1% and 98.9% at $k{=}1$, and $k{=}20$ order decisions remain above 72% in both settings. The original Pile-style pretraining-domain and diffusion experiments remain in the main body as the cross-model validation of the same primitive. Second, sequence and large-$N$ experiments show that pairwise bracket edges compose: the method recovers the best of all 3! schedules 87.5% of the time, places it in the top two 96.0% of the time, ranks all 85 Stack programming-language source domains for a Python target in the 99th sampled percentile, and reaches the 99.0–99.6th sampled percentile on 56 MMLU subjects. Third, long-horizon and wall-clock studies show that the planner remains useful when brute-force order testing becomes expensive: over 1,020 Llama trials, accuracy is 81.5% at $k{=}20$ and 65.3% at $k{=}50$, while planning is already faster than trying both orders around $k \approx 3$.

**Contributions.** We summarize our contributions as follows.

1. **A commutator theory of transfer order.** We connect the order-dependent target-loss difference to the directional bracket score $\sigma_{AB}^{(E)} = \langle g_E, b_{AB} \rangle$ and show that its sign predicts whether $A{\to}B$ or $B{\to}A$ yields lower

target loss.

2. **A deployable pairwise planner.** We introduce a drift-matched Trotter estimator for $\sigma$ using gradients and Hessian-vector products in a chosen trainable parameter subset, together with stakes and confidence scores.

3. **Lie-Bracket Tournaments for many-domain scheduling.** We show that the same bracket primitive defines a weighted tournament over source domains, enabling curriculum ranking without evaluating $N!$ permutations. The scalable Borda/row-sum variant deliberately uses a shared $\theta_0$ target-gradient reference, not the pair-specific Trotter reference, which yields $N$-independent edges, one HVP per source, linear score computation, and an $O(N \log N)$ sort.

4. **Validation across post-training and domain adaptation.** We evaluate instruction-SFT, DPO/offline preference optimization, next-token pretraining-domain adaptation, diffusion adaptation, long-horizon training up to $k{=}50$, large-$N$ scheduling up to $N{=}85$, wall-clock planning cost, and stateful-optimizer behavior.

5. **Theory-driven step-size selection.** We develop an automated $\eta$-autopilot that selects step sizes from pilot data by balancing detectability against higher-order sign-stability constraints, avoiding manual per-model tuning.

## 2. Preliminaries

### 2.1. Notation and Setup

Let $L_D(\theta)$ denote the expected training loss on dataset/domain $D$ for parameters $\theta \in \mathbb{R}^p$. We write gradients and Hessians as

$$g_D(\theta) = \nabla_\theta L_D(\theta), \qquad H_D(\theta) = \nabla_\theta^2 L_D(\theta). \quad (3)$$

We consider three domains: two *source* domains $A$, $B$ and one *target* domain $E$. For each domain, we compute gradients and Hessians on training batches, and evaluate final losses $L_E(\theta_{AB})$ vs. $L_E(\theta_{BA})$ on held-out evaluation batches.

We model a single gradient step on domain $D$ with step size $\eta > 0$ as the operator

$$U_D(\theta) := \theta - \eta\, g_D(\theta). \quad (4)$$

Although practical fine-tuning runs many optimizer steps (often with adaptive methods), we use $U_D$ as a local surrogate for a short fine-tuning phase. We evaluate gradients and HVPs at a reference point and use an effective step size $\eta$ (the per-step learning rate in our experiments). We apply the commutator logic locally and empirically verify that the

predictor remains accurate beyond the infinitesimal regime. Throughout, we use $k$ to denote the number of gradient steps per domain (so $k=1$ means one step on $A$ then one step on $B$; $k=20$ means 20 steps on each, 40 total per ordering). Given an initialization $\theta_0$, the two-step compositions are

$$\theta_{AB} := U_B(U_A(\theta_0)), \qquad \theta_{BA} := U_A(U_B(\theta_0)). \quad (5)$$

Our object of interest is the *order effect on the target loss*

$$\Delta_E(A, B) := L_E(\theta_{AB}) - L_E(\theta_{BA}). \quad (6)$$

---

**Problem Definition.** *Input:* Source domains $A, B$, target domain $E$, initialization $\theta_0$, step size $\eta$, trainable parameter subset $S$. *Output:* Predict $\arg\min\{L_E(\theta_{AB}), L_E(\theta_{BA})\}$, stakes score $s$, and confidence score $c$.

---

### 2.2. Commutators and Lie Brackets of Update Operators

Order dependence arises because the update maps $U_A, U_B$ do not commute: $\theta_{AB} \neq \theta_{BA}$ in general. For gradient descent steps, this non-commutativity admits a controlled second-order expansion that isolates a single geometric object.

**Lemma 2.1** (Order dependence of two gradient steps). *Let $L_A$ and $L_B$ be three times continuously differentiable in a neighborhood of $\theta_0$. Assume their Hessians are locally Lipschitz near $\theta_0$;[1] that is, there exist constants $M_3 > 0$ and $R > 0$ such that for all $\theta$ with $\|\theta - \theta_0\| \leq R$,*

$$\|H_A(\theta) - H_A(\theta_0)\| \leq M_3\|\theta - \theta_0\|,$$
$$\|H_B(\theta) - H_B(\theta_0)\| \leq M_3\|\theta - \theta_0\|.$$

*Then for sufficiently small $\eta$ (so that intermediate iterates remain within the ball),*

$$\theta_{AB} - \theta_{BA} = \eta^2 \Big(H_B(\theta_0)\, g_A(\theta_0) - H_A(\theta_0)\, g_B(\theta_0)\Big)$$
$$+ r_{AB}, \quad (7)$$

*where the remainder satisfies $\|r_{AB}\| \leq C\, \eta^3$ for a constant $C$ depending only on local smoothness (e.g. $M_3$) and the local gradient norms at $\theta_0$.*

*Remark* 2.2 (Projected updates). All statements in this section hold verbatim for projected updates $U_{D,S}(\theta) = \theta - \eta P_S g_D(\theta)$, where $P_S$ is a fixed linear projection (e.g., coordinate projection onto a chosen parameter block), by replacing gradients with $P_S g_D$ and Hessians with $P_S H_D P_S$ (equivalently, working in coordinates on $S$).

---

[1]Equivalently, the third derivatives are bounded on a neighborhood. This is the standard smoothness condition needed to make the $O(\eta^3)$ remainder explicit.

**Interpretation.** Define the *bracket vector* (at $\theta_0$)

$$b_{AB}(\theta_0) := H_B(\theta_0)\, g_A(\theta_0) - H_A(\theta_0)\, g_B(\theta_0). \quad (8)$$

Then Lemma 2.1 states that the leading-order driver of order dependence is $\theta_{AB} - \theta_{BA} \approx \eta^2 b_{AB}(\theta_0)$.

**Connection to Lie brackets.** Let $f_A(\theta) := -g_A(\theta)$ and $f_B(\theta) := -g_B(\theta)$ be the gradient-flow vector fields. Their Lie bracket is

$$[f_A, f_B](\theta) := \big(\nabla f_B(\theta)\big)f_A(\theta) - \big(\nabla f_A(\theta)\big)f_B(\theta)$$
$$= H_B(\theta)\, g_A(\theta) - H_A(\theta)\, g_B(\theta),$$

so $b_{AB}(\theta_0)$ equals $[f_A, f_B](\theta_0)$ under our sign convention. Thus the same commutator term that appears in the Baker–Campbell–Hausdorff (BCH) expansion (Baker, 1905; Campbell, 1896; Hausdorff, 1906) for continuous-time flows also governs the order sensitivity of discrete gradient steps (to second order).

Appendix A.1 gives the proof with explicit constants. The main text uses only the second-order scaling and the identification of the leading commutator term.

### 2.3. From Parameter Commutators to Target Loss Differences

Let $E$ denote a *target* domain. The order effect on $E$ is

$$\Delta_E(A, B) := L_E(\theta_{AB}) - L_E(\theta_{BA}). \quad (9)$$

**Proposition 2.3** (Directional bracket approximation of target loss difference). *Assume $L_E$ is twice continuously differentiable near $\theta_0$ with a locally bounded Hessian, and that the assumptions of Lemma 2.1 hold. Then, as $\eta \to 0$,*

$$\Delta_E(A, B) = \eta^2 \sigma_{AB}^{(E)}(\theta_0) + O(\eta^3),$$
$$\sigma_{AB}^{(E)}(\theta_0) := \langle g_E(\theta_0), b_{AB}(\theta_0)\rangle. \quad (10)$$

**Sign convention (the decision rule).** Since $\Delta_E(A, B) < 0$ means $A \to B$ yields *lower* target loss than $B \to A$, Proposition 2.3 motivates the predictor

$$\widehat{\text{order}}(A, B; E) = \begin{cases} A \to B & \text{if } \hat{\sigma}_{AB}^{(E)} < 0, \\ B \to A & \text{otherwise.} \end{cases} \quad (11)$$

**Normalized score (optional but useful).** We reserve $\sigma$ for the unnormalized directional score; $\omega$ denotes an optional scale-free cosine. In practice it is often convenient to report the scale-free cosine score

$$\omega_{AB}^{(E)}(\theta) := \frac{\langle g_E(\theta), b_{AB}(\theta_0)\rangle}{\|g_E(\theta)\|\, \|b_{AB}(\theta_0)\| + \varepsilon}, \quad (12)$$

whose sign matches the numerator $\langle g_E(\theta), b_{AB}(\theta_0)\rangle$. In particular, at $\theta = \theta_0$ we have $\text{sign}(\omega_{AB}^{(E)}(\theta_0)) = \text{sign}(\sigma_{AB}^{(E)}(\theta_0))$. The unnormalized $\sigma$ is the quantity directly appearing in the asymptotic expansion (10).

**Stakes and confidence.** Proposition 2.3 suggests that the *magnitude* of the effect is $|\Delta_E(A,B)| \approx \eta^2 |\sigma_{AB}^{(E)}|$. Accordingly, we define the predicted stakes score

$$s_{AB}^{(E)} := \eta^2 \, |\hat{\sigma}_{AB}^{(E)}|. \tag{13}$$

In experiments, accuracy increases as we restrict to pairs with large $|\hat{\omega}|$ (strong alignment between $g_E$ and the bracket), which are empirically much more stable across seeds (Table 2).

*Proof sketch.* By the mean value theorem,

$$\Delta_E(A,B) = \langle g_E(\theta^\star), \theta_{AB} - \theta_{BA} \rangle,$$
$$\theta^\star \in [\theta_{BA}, \theta_{AB}].$$

Using Lemma 2.1, $\theta_{AB} - \theta_{BA} = \eta^2 b_{AB}(\theta_0) + O(\eta^3)$. Moreover, $\theta^\star = \theta_0 + O(\eta)$ (since both $\theta_{AB}$ and $\theta_{BA}$ are $O(\eta)$ away from $\theta_0$), so $g_E(\theta^\star) = g_E(\theta_0) + O(\eta)$ under bounded $H_E$. Multiplying gives $\Delta_E(A,B) = \eta^2 \langle g_E(\theta_0), b_{AB}(\theta_0) \rangle + O(\eta^3)$. A complete proof is given in Appendix A.2.

## 2.4. Reference-Point Estimators

Proposition 2.3 evaluates the directional bracket using $g_E(\theta_0)$. However, the mean value theorem implies there exists $\theta^\star \in [\theta_{BA}, \theta_{AB}]$ such that

$$\Delta_E(A,B) = \langle g_E(\theta^\star), \theta_{AB} - \theta_{BA} \rangle,$$

so the relevant target gradient is generally evaluated at an *intermediate* point between the two endpoints. A key observation is that both orders share a common first-order drift $-\eta(g_A + g_B)$ from $\theta_0$, so $[\theta_{BA}, \theta_{AB}]$ lies within $O(\eta^2)$ of a single reference point. Evaluating $g_E$ there approximates $g_E(\theta^\star)$ to second order.

Define the reference point

$$\theta_{\text{ref}} := \theta_0 - \eta\big(g_A(\theta_0) + g_B(\theta_0)\big). \tag{14}$$

**Lemma 2.4** (Shared first-order drift). *Assume the conditions of Lemma 2.1. Then the two-step endpoints satisfy*

$$\theta_{AB} = \theta_{\text{ref}} + \eta^2 H_B(\theta_0) g_A(\theta_0) + O(\eta^3),$$
$$\theta_{BA} = \theta_{\text{ref}} + \eta^2 H_A(\theta_0) g_B(\theta_0) + O(\eta^3).$$

*In particular, $\|\theta_{AB} - \theta_{\text{ref}}\| \vee \|\theta_{BA} - \theta_{\text{ref}}\| = O(\eta^2)$.*

**Proposition 2.5** (Drift-matched target gradient). *Assume the conditions of Lemma 2.1 and that $g_E$ is locally $M_{2,E}$-Lipschitz (i.e., $\|H_E(\theta)\| \le M_{2,E}$ in a neighborhood of $\theta_0$). Let $d := \theta_{AB} - \theta_{BA}$. Then there exists $\theta^\star \in [\theta_{BA}, \theta_{AB}]$ such that $\Delta_E(A,B) = \langle g_E(\theta^\star), d \rangle$, and moreover $\|\theta^\star - \theta_{\text{ref}}\| = O(\eta^2)$. Consequently,*

$$\big|\Delta_E(A,B) - \langle g_E(\theta_{\text{ref}}), d \rangle\big| \le \|g_E(\theta^\star) - g_E(\theta_{\text{ref}})\| \, \|d\|$$
$$\le M_{2,E} \, \|\theta^\star - \theta_{\text{ref}}\| \, \|d\|$$
$$= O(\eta^4).$$

*Remark* 2.6 (What Trotter is correcting). Proposition 2.5 isolates the *target-gradient drift* term: evaluating $g_E$ at $\theta_0$ instead of $\theta_{\text{ref}}$ incurs an error of order $O(\eta) \times O(\eta^2) = O(\eta^3)$ when paired with the commutator displacement. In contrast, $\theta^\star$ is $O(\eta^2)$-close to $\theta_{\text{ref}}$, so replacing $g_E(\theta^\star)$ by $g_E(\theta_{\text{ref}})$ contributes only $O(\eta^4)$. The remaining $O(\eta^3)$ approximation error in the deployable score is then dominated by approximating $d$ by the second-order bracket $\eta^2 b_{AB}(\theta_0)$ (Lemma 2.1), rather than by drift of $g_E$ along the shared first-order trajectory.

We then define three estimators of $\sigma_{AB}^{(E)}$ by choosing different approximations to the *average* target gradient along this shared drift:

**Base (single-point).** We define

$$\hat{\sigma}_{AB}^{(E)}(\text{base}) := \langle g_E(\theta_0), b_{AB}(\theta_0) \rangle. \tag{15}$$

**Trotter reference (single-point at $\theta_{\text{ref}}$) (Trotter, 1959).** We define

$$\hat{\sigma}_{AB}^{(E)}(\text{trotter}) := \langle g_E(\theta_{\text{ref}}), b_{AB}(\theta_0) \rangle. \tag{16}$$

**Trapezoid reference (two-point average; ablation).** We define

$$\hat{\sigma}_{AB}^{(E)}(\text{trap\_local}) := \frac{1}{2} \langle g_E(\theta_0) + g_E(\theta_{\text{ref}}), b_{AB}(\theta_0) \rangle. \tag{17}$$

All three estimators are *deployable*: they do not require running both orders to obtain $\theta_{AB}$ and $\theta_{BA}$. They differ only in where they evaluate the target gradient $g_E$. Empirically, performance varies by model and operating point: the single-point Trotter estimator is the most reliable in the harder settings (e.g., SmolLM3), while trapezoid averaging can help in some models. We also evaluate *oracle* estimators that use the true endpoints $\theta_{AB}, \theta_{BA}$ (see Appendix B); these serve as upper bounds but are not deployable for order selection.

**Stochastic approximation.** In practice we use minibatch gradients; all reported results use the same minibatch draws for both orders to isolate order-dependence from sampling noise.

## 2.5. Projected Computation in a Trainable Subspace

Our estimator never forms a Hessian matrix. It only requires two Hessian–vector products (HVPs), $H_B g_A$ and $H_A g_B$, computed with respect to the parameters we actually fine-tune. We therefore restrict all computations to a *fixed trainable parameter subset* (a coordinate subspace)—the parameters that will be updated—and treat all other parameters as frozen.

For LLMs, we select only the *final transformer layer's* output projection (`o_proj`) and feed-forward down projection (`down_proj`), which is ~1–2% of parameters. We choose post-attention linear layers so HVPs avoid higher-order differentiation through attention. For the diffusion model, we select all convolutional layers (excluding attention and normalization), comprising the majority of UNet parameters. These choices align with standard fine-tuning practice for each architecture: final-layer tuning for LLMs is a simple parameter-efficient regime (comparable in spirit to adapters/LoRA), while full-conv tuning for UNets matches typical diffusion domain adaptation.

Let $S \subseteq \mathbb{R}^p$ denote the coordinate subspace of trainable parameters and $P_S$ the corresponding fixed linear projection. We compute projected gradients and Hessian-vector products: $g_D^S := P_S g_D$, $H_D^S v := P_S(H_D(P_S v))$. The HVP uses Pearlmutter's trick (Pearlmutter, 1994): $Hv = \nabla(\nabla L \cdot v)$, requiring only a standard double backward rather than explicit Hessian materialization. HVPs can also be computed in a distributed manner for large Transformer models (Granziol, 2025). In general, an HVP costs a small constant factor more than a gradient; in our subspace-restricted implementation, each HVP is approximately 2× a gradient on the selected parameters. We validated that exact second-order information is important: finite-difference HVP approximation degrades accuracy by 4–12pp (Appendix D.8), confirming that precise curvature is necessary to capture the commutator geometry. The bracket vector is then $b_{AB}^S := H_B^S g_A^S - H_A^S g_B^S$, and all scores $\sigma$ are computed using these projected quantities.

This subspace restriction makes the planner naturally compatible with parameter-efficient fine-tuning methods.

**Scope of validation.** In our experiments, we train and evaluate within the same parameter subset: both the ground-truth order effects (from running both orders) and the predicted commutator scores are computed on the selected trainable parameters. This validates order prediction for parameter-efficient fine-tuning scenarios, where practitioners need to choose an order for adapter/LoRA/final-layer training. To test robustness beyond the tiny final-layer subset, we also run an expanded multi-layer validation on Llama-3.2-1B tuning the last 8 transformer layers (layers 8–15; `o_proj`+`down_proj` in each), achieving 825/1020 sign accuracy (80.9%) in the larger multi-seed evaluation (Appendix D.6). A small eager-attention ablation further suggests that excluding attention input projections is mainly a systems constraint rather than a theoretical one: on the 50-triple probe, post-attention layers alone and an eager-attention subset including `q_proj`, `k_proj`, `v_proj`, `o_proj`, and `down_proj` both achieve 92.0% sign accuracy.

---

**Algorithm 1** Transfer-Order Planner (Trotter / endpoint reference point)

---

**Require:** Sources $A, B$, target $E$, params $\theta_0$, step size $\eta$, subset $S$
**Ensure:** Predicted order, stakes $s$, confidence $c$
1: Compute $g_A \leftarrow P_S \nabla L_A(\theta_0)$
2: Compute $g_B \leftarrow P_S \nabla L_B(\theta_0)$
3: *// Hessian-vector products*
4: $H_B g_A \leftarrow P_S \nabla^2 L_B(\theta_0) \cdot g_A$
5: $H_A g_B \leftarrow P_S \nabla^2 L_A(\theta_0) \cdot g_B$
6: $b \leftarrow H_B g_A - H_A g_B$     *// Bracket vector*
7: *// Reference point for shared drift*
8: $\theta_{\text{ref}} \leftarrow \theta_0 - \eta(g_A + g_B)$
9: $g_E^{(\text{ref})} \leftarrow P_S \nabla L_E(\theta_{\text{ref}})$
10: $\hat{\sigma} \leftarrow \langle g_E^{(\text{ref})}, b \rangle$     *// Trotter score*
11: $s \leftarrow \eta^2 |\hat{\sigma}|$   *// Stakes (predicted effect magnitude)*
12: $c \leftarrow |\hat{\sigma}|/(\|g_E^{(\text{ref})}\|\|b\|)$   *// Confidence (alignment $|\hat{\omega}|$)*
13: **if** $\hat{\sigma} > 0$ **then**
14:     **return** $B \rightarrow A, s, c$
15: **else**
16:     **return** $A \rightarrow B, s, c$
17: **end if**

---

## 3. Method: Transfer-Order Planner

We now present the complete transfer-order planning algorithm. Given source domains $A, B$, target domain $E$, initial parameters $\theta_0$, step size $\eta$, and trainable parameter subset $S$, our planner outputs a predicted optimal order, a stakes score, and a confidence score.

**Compute cost.** Let $C_g$ denote one gradient evaluation and $C_h$ one Hessian-vector product (HVP) in the chosen trainable subspace, with $\rho = C_h/C_g \approx 2$ in our implementation. Algorithm 1 computes three gradients and two HVPs, so $C_{\text{plan}} = (3 + 2\rho)C_g$ for one pair. Trying both $k$-step orders costs $C_{\text{both}}(k) = 4kC_g$ before final evaluation. Thus the planner is cheaper for decision-only order selection when $k > (3 + 2\rho)/4$ and cheaper end-to-end when $k > (3 + 2\rho)/2$. With $\rho \approx 2$, this predicts a practical crossover around $k \approx 3$–$4$; the wall-clock measurements in Appendix E.7 match this prediction.

### 3.1. Lie-Bracket Tournaments for $N$ Sources

The pairwise planner induces a many-domain scheduler by treating every pairwise preference as an edge in a weighted tournament. Fix a target domain $E$ and candidate source set $\mathcal{D} = \{D_1, \ldots, D_N\}$. Let $g_i := g_{D_i}(\theta_0)$ and $H_i := H_{D_i}(\theta_0)$, and choose a shared target-gradient reference $g_E^{\text{ref}}$. For an ordered pair $(i, j)$, the pairwise bracket score is

$$\sigma_{ij}^{(E)} := \langle g_E^{\text{ref}}, H_j g_i - H_i g_j \rangle. \tag{18}$$

Because $\sigma_{ij}^{(E)} < 0$ predicts $D_i \to D_j$ and $\sigma_{ij}^{(E)} > 0$ predicts $D_j \to D_i$, it is convenient to define the tournament edge

$$W_{ij}^{(E)} := -\sigma_{ij}^{(E)} = \langle g_E^{\mathrm{ref}}, H_i g_j - H_j g_i \rangle, \qquad (19)$$

so $W_{ij}^{(E)} > 0$ means that $D_i$ should precede $D_j$ for target $E$. The matrix $W^{(E)}$ is skew-symmetric up to stochastic estimation error and defines a weighted tournament over the source domains. We rank this tournament by the classical Borda/row-sum score (de Borda, 1781; Brandt et al., 2016):

$$r_i^{(E)} := \sum_{j \neq i} W_{ij}^{(E)}, \qquad (20)$$

placing larger $r_i^{(E)}$ earlier. For small $N$, we can also score complete permutations by the same pairwise surrogate $S(\pi) = \sum_{a<b} W_{\pi(a),\pi(b)}^{(E)}$ and compare all schedules.

The main computational identity is Hessian symmetry. Define

$$u_i^{(E)} := H_i g_E^{\mathrm{ref}}. \qquad (21)$$

Then

$$W_{ij}^{(E)} = \langle g_j, u_i^{(E)} \rangle - \langle g_i, u_j^{(E)} \rangle. \qquad (22)$$

Thus all pairwise edges can be filled from $g_i$ and $u_i^{(E)}$ using dot products. For Borda/row-sum ranking, even the $O(N^2)$ edge matrix is unnecessary: if

$$G := \sum_j g_j, \qquad U^{(E)} := \sum_j u_j^{(E)}, \qquad (23)$$

then self-terms cancel and

$$\begin{aligned} r_i^{(E)} &= \sum_j \langle g_j, u_i^{(E)} \rangle - \sum_j \langle g_i, u_j^{(E)} \rangle \\ &= \langle G, u_i^{(E)} \rangle - \langle g_i, U^{(E)} \rangle. \end{aligned} \qquad (24)$$

For a fixed target and shared target-gradient reference, Borda score computation therefore needs $N$ source gradients, $N$ HVPs $u_i^{(E)} = H_i g_E^{\mathrm{ref}}$, two vector sums, and $2N$ dot products, followed by an $O(N \log N)$ sort. In all large-$N$ tournament experiments we set the shared target reference to the initialization, $\theta_E^{\mathrm{ref}} = \theta_0$, so $g_E^{\mathrm{ref}} = g_E(\theta_0)$. This choice is deliberate: each edge $W_{ij}$ then depends only on domains $i, j$ and target $E$, making existing edges $N$-independent and enabling the row-sum factorization in Eq. (24). A drift-matched reference would depend on the chosen pair or on the whole source set, which would break this factorization and reintroduce $O(N^2)$ reference scoring. The reduction is exact for Borda/row-sum aggregation with this shared reference; exact tournament optimization, local-search variants, or pair-specific Trotter references require materializing or querying the pairwise matrix.

Algorithm 2 is a local second-order surrogate for sequence-level performance. For $N{=}3$, we can evaluate the complete

---

**Algorithm 2** Lie-Bracket Tournament Ranking

**Require:** Sources $D_1, \ldots, D_N$, target $E$, params $\theta_0$, subset $S$, shared target reference $\theta_E^{\mathrm{ref}}$
**Ensure:** Curriculum ranking of the $N$ sources
1: Compute $g_E^{\mathrm{ref}} \leftarrow P_S \nabla L_E(\theta_E^{\mathrm{ref}})$
2: **for** $i = 1, \ldots, N$ **do**
3:    $g_i \leftarrow P_S \nabla L_{D_i}(\theta_0)$
4:    $u_i \leftarrow P_S \nabla^2 L_{D_i}(\theta_0) \, g_E^{\mathrm{ref}}$    *// one HVP per source*
5: **end for**
6: $G \leftarrow \sum_i g_i, \quad U \leftarrow \sum_i u_i$
7: **for** $i = 1, \ldots, N$ **do**
8:    $r_i \leftarrow \langle G, u_i \rangle - \langle g_i, U \rangle$    *// Borda/row-sum score*
9: **end for**
10: **return** sources sorted by decreasing $r_i$

---

surrogate over all 3! permutations; for large $N$, Borda/row-sum ranking gives a scalable tournament order. The large-$N$ tournament and the pairwise planner therefore use the same bracket vector but different target-gradient references: Algorithm 1 uses a drift-matched Trotter point for the most accurate two-domain sign decision, while Algorithm 2 uses the shared $\theta_0$ reference to obtain $N$-independent edges and linear row-sum scoring. Drift matching is an $O(\eta^3)$ correction to an individual pairwise loss difference; Borda aggregation asks for a robust ranking over many local edges, and the reported tournament experiments test whether the cheaper shared-reference edges remain informative enough to beat random curricula and first-order baselines.

**Confidence gating.** We distinguish between *stakes* (how much the order matters) and *confidence* (how reliable the sign prediction is). The stakes score $s = \eta^2 |\hat{\sigma}|$ estimates the effect magnitude $|\Delta_E|$. For confidence we use the scale-free alignment score $|\hat{\omega}|$, which is large when $g_E$ and the bracket are nearly collinear and small when they are nearly orthogonal. High $|\hat{\omega}|$ cases are empirically much more stable across seeds.

1. **Effect magnitude:** Large $s$ indicates the order choice has substantial impact on target loss.

2. **Prediction reliability:** Large $|\hat{\omega}|$ indicates strong directional alignment between $g_E$ and the bracket, yielding reliable sign prediction.

We recommend deploying the planner when $|\hat{\omega}|$ exceeds a coverage threshold; low-confidence cases have small effects where either order suffices. In Table 2, Acc@25% reports *trotter* accuracy on the top quartile ranked by actual impact $|\Delta_E|$.

**Step-size selection ($\eta$-autopilot).** The reference point $\theta_{\mathrm{ref}}$ and the predicted stakes $s = \eta^2 |\hat{\sigma}|$ depend on the step size

$\eta$, which must be large enough for order effects to be detectable but small enough that the score's sign is stable beyond the second-order expansion. For practitioners who prefer simplicity, a coarse grid search over $\eta$ also works well (Appendix D.2). However, to avoid per-model tuning, we develop a theory-driven autopilot that selects $\eta$ automatically from $\sim 40$ pilot triples ($\sim 10$ minutes) by combining a detectability floor with sign-validity ceilings. Let $S_q$ be a high quantile of $|\hat{\sigma}|$ over pilot triples and let $\sigma_{\text{eff}}$ denote a conservative standard error (in units of loss) for the paired difference $\Delta_E(A, B)$, estimated from repeated AB/BA evaluations at a small reference step $\eta_{\text{ref}}$. Modeling $\Delta_E(A, B) \approx \eta^2 \sigma + \varepsilon$ with $\text{SE}(\varepsilon) \approx \sigma_{\text{eff}}$, we require a typical high-signal effect $\eta^2 S_q$ to exceed a noise threshold $z\,\sigma_{\text{eff}}$, yielding

$$\eta^2 S_q \geq z\,\sigma_{\text{eff}} \quad \Longrightarrow \quad \eta_{\min} := \sqrt{\frac{z\,\sigma_{\text{eff}}}{S_q}}. \qquad (25)$$

We then choose $\eta$ using the implementation-matched cube-root policy in Appendix D.1. Let $\eta_{\text{sign}}$ be the aggregated sign-validity ceiling and let $\eta_{\text{tradeoff}}$ be a BCH-based ceiling with modest headroom. The selected step is always clamped by sign-validity; in the LLM runs, an entanglement-aware N-finder is active and the slack-ratio branch uses cube-root interpolation rather than collapsing to the bare detectability floor. When pilot triples underestimate evaluation-scale effects, the implemented floor applies the corresponding correction $\eta_{\min} \leftarrow \eta_{\min}/\sqrt{\hat{N}_{\text{eff}}}$ before the regime gates. Full code-matched details are in Appendix D.1.

# 4. Experiments

## 4.1. Experimental Setup

**Models and training regimes.** We evaluate the same bracket primitive in four regimes. First, for instruction-SFT we fine-tune Qwen2.5-1.5B on Dolly-style task categories treated as domains (Conover et al., 2023). Second, for DPO we treat UltraFeedback preference subsets as source domains in an offline preference-optimization setting (Cui et al., 2023; Rafailov et al., 2023). Third, for next-token pretraining-domain adaptation we evaluate Qwen2.5-1.5B (Yang et al., 2024), Llama-3.2-1B (Meta AI, 2024), Llama-3.1-8B (Grattafiori et al., 2024), and SmolLM3-3B (Bakouch et al., 2025). Fourth, we evaluate a DDPM UNet (Ho et al., 2020) on class-conditioned diffusion adaptation. The SFT and DPO settings are the most direct post-training applications; the original Pile-style pretraining-domain and diffusion settings test that the same geometric primitive also covers the earlier domain-adaptation case and architectural breadth.

**Domains.** For the original pairwise LLM experiments, we use The Pile dataset (Gao et al., 2020) via the

ola13/small-the_pile Hugging Face subset. After filtering domains with fewer than 40 available samples, 17 domains remain: Pile-CC, OpenWebText2, PubMed Abstracts, StackExchange, Github, Wikipedia (en), USPTO Backgrounds, PubMed Central, FreeLaw, NIH ExPorter, ArXiv, DM Mathematics, HackerNews, Enron Emails, OpenSubtitles, Books3, and YoutubeSubtitles. For each target $E$, we sample 12 random source pairs $(A, B)$ with $A \neq B$ and $A, B \neq E$, yielding $17 \times 12 = 204$ triples per experiment. For diffusion, CIFAR-10 classes are domains; fixing a target class gives $\binom{9}{2} = 36$ source pairs. Large-$N$ scheduling uses MMLU subjects (Hendrycks et al., 2021), Stack/programming-language domains, and Dolly task categories.

**Numerical precision and metrics.** HVPs are computed in float32 for stability: Llama-3.2-1B and the diffusion model run in full float32, while larger LLMs keep the model in bfloat16 but cast the selected trainable parameter subset to float32 for HVP computation. We report sign accuracy, regret $L_E(\theta_{\text{chosen}}) - \min\{L_E(\theta_{AB}), L_E(\theta_{BA})\}$, BCH validation $\cos(\theta_{AB} - \theta_{BA}, \eta^2 b)$, and for many-domain schedules percentile against sampled random curricula. Step sizes in the pairwise table are selected by the theory-driven $\eta$-autopilot from a small disjoint pilot set.

## 4.2. Post-Training: Instruction-SFT and DPO

A central broadening beyond the original domain-adaptation setting is post-training. We test whether the same bracket score predicts order effects for instruction supervised fine-tuning and preference optimization, using Qwen2.5-1.5B in float32 and held-out target losses. These experiments are important because SFT and DPO use different losses and sharper task/domain boundaries than Pile-style next-token domain adaptation.

**Instruction-SFT.** We use Dolly-15k task categories (Conover et al., 2023) as source and target domains. Across 480 triples over five seeds, the pairwise bracket planner achieves 471/480 = 98.1% sign accuracy at $k{=}1$, mean Spearman 0.940, and 96.3% recovered fraction. When the same $k{=}1$ predictor is held fixed and ground truth is evaluated after $k{=}20$ SFT steps per domain, accuracy remains 351/480 = 73.1% with 58.0% regret reduction.

**DPO / offline preference optimization.** We use Ultra-Feedback source domains (Cui et al., 2023) and the DPO loss (Rafailov et al., 2023), excluding within-flan_v2 pairs that share provenance and create near-degenerate brackets. Across 356 triples over four seeds, the pairwise bracket planner reaches 352/356 = 98.9% sign accuracy at $k{=}1$ and 97.8% recovered fraction. At $k{=}20$, using the same fixed initial predictor, accuracy is 257/356 = 72.2% with

*Table 1.* Post-training pairwise prediction. The $k{=}20$ columns evaluate 20 steps per source while keeping the initial bracket predictor fixed.

| Setting | $k{=}1$ accuracy | Rank quality | $k{=}20$ accuracy | Regret red. |
|---|---|---|---|---|
| Instruction-SFT | 471/480 = 98.1% | Spearman 0.940 | 351/480 = 73.1% | 58.0% |
| DPO | 352/356 = 98.9% | recovered 97.8% | 257/356 = 72.2% | 54.7% |

*Table 2.* Original pretraining-domain and diffusion validation. LLM rows use Pile-style next-token domains and DDPM uses class-conditioned diffusion domains; accuracies are mean $\pm$ std over 5 seeds. Acc@25% is Trotter accuracy on the 25% highest-impact decisions ranked by $|\Delta_E|$.

| Model | Trotter Acc. | Cosine Acc. | Acc.@25% | $\eta$ |
|---|---|---|---|---|
| Qwen2.5-1.5B | 87.7%$\pm$1.8 | 61.8% | 95.7%$\pm$1.5 | 0.00203 |
| Llama-3.2-1B | 92.0%$\pm$2.2 | 77.9% | 82.4%$\pm$7.9 | 0.00105 |
| Llama-3.1-8B | 82.4%$\pm$2.4 | 67.2% | 94.9%$\pm$1.6 | 0.00133 |
| SmolLM3-3B | 87.2%$\pm$2.6 | 61.3% | 86.3%$\pm$2.5 | 0.00178 |
| DDPM UNet | 91.1%$\pm$3.7 | 88.9% | 100.0%$\pm$0.0 | 0.0906 |

*Table 3.* Long-horizon generalization on Llama-3.2-1B over 1,020 trials. The $k{=}1$ predictor is computed once at $\theta_0$; ground truth uses $k$ steps per domain.

| Horizon | Correct/total | Accuracy | Regret red. |
|---|---|---|---|
| $k{=}1$ | 949/1020 | 93.0% | 64.0% |
| $k{=}5$ | 962/1020 | 94.3% | 97.4% |
| $k{=}10$ | 912/1020 | 89.4% | 93.1% |
| $k{=}20$ | 831/1020 | 81.5% | 86.3% |
| $k{=}50$ | 666/1020 | 65.3% | 54.7% |

*Table 4.* Large-$N$ tournament scheduling. Percentiles are relative to 500 sampled random curricula; "Grad-norm" is descending gradient-norm ordering.

| Dataset / target | $N, k$ | Bracket pct. | Grad-norm pct. |
|---|---|---|---|
| MMLU subjects | 56, 1 | 99.0–99.6 | $< 1$ |
| MMLU subjects | 30, 5 | 89–96 | 2–22 |
| Stack / Python | 85, 1 | 99 | 69 |
| Stack / Python | 10, 5 | 86 | 47 |
| Dolly / summarization | 7, 5 | 99.8 | 49.9 |

54.7% regret reduction. We describe this as offline preference optimization rather than online RLHF: it tests whether the bracket transfers to preference landscapes, while online sampling and policy-optimization effects are left for future work.

### 4.3. Pairwise Pretraining-Domain Adaptation and Long Horizons

The tournament construction depends on the pairwise bracket score being reliable. Table 2 preserves the original cross-architecture validation on Pile-style next-token pretraining-domain adaptation and diffusion: the Trotter planner achieves strong overall sign accuracy and improves sharply on high-impact decisions across LLM families and diffusion. This validates the pairwise score as the primitive used by the tournament.

The theory is local, so a key empirical question is whether the score survives beyond one gradient step. Table 3 uses 1,020 Llama-3.2-1B trials, computes the predictor once at $\theta_0$, and evaluates ground truth after $k$ SGD steps per domain. Accuracy declines gradually with horizon while remaining above random: 81.5% at $k{=}20$ and 65.3% at $k{=}50$ (100 updates total), with regret reduction still above random.

### 4.4. Many-Domain Scheduling by Lie-Bracket Tournament

The tournament layer asks a different question from pairwise sign prediction: can local pairwise geometry produce a useful complete curriculum? For $N{=}3$, the answer can be tested exactly because all 3! schedules are evaluated for ground truth. The complete pairwise surrogate recovers the best schedule in 175/200 cases (87.5%), places the best schedule in the top two in 192/200 cases (96.0%), obtains

mean Spearman 0.929, and reduces regret by 93.5%. The scalable Borda/row-sum variant, which is the one used for large $N$, still recovers the best $N{=}3$ order in 79.0% of cases, far above the 16.7% random top-1 rate. Only 1.5% of quadruples show cyclic intransitivity, and those cycles concentrate near indifferent pairs.

For larger $N$, exhaustive evaluation of all schedules is impossible, so we compare the tournament-selected order to 500 sampled random curricula and to gradient-norm baselines. The main metric is therefore a *sampled percentile*, not top-1 accuracy over all $N!$ schedules. On MMLU subject curricula, the bracket tournament reaches the 99.0–99.6th sampled percentile at $N{=}56, k{=}1$ across the reported seeds, while the reported descending gradient-norm baseline is below the first percentile. We interpret this as a limitation of magnitude-only scheduling rather than as a claim about every possible norm orientation: gradient norm is not target-conditioned, whereas the bracket edge uses the target projection of non-commutativity. On Stack programming-language curricula, the $N{=}85, k{=}1$ bracket order reaches the 99th sampled percentile for a Python target, versus the 69th percentile for the descending gradient-norm baseline. The same construction also performs strongly at intermediate sizes: MMLU $N{=}30, k{=}5$ reaches the 89–96th sampled percentile while gradient-norm ordering reaches only 2–22, Stack $N{=}10, k{=}5$ reaches the 86th versus 47, and a Dolly summarization target reaches the 99.8th versus 49.9.

The qualitative message is that the pairwise bracket is an $N$-independent primitive: each edge asks the same local geometric question, and Algorithm 2 turns the resulting tournament into a curriculum without enumerating permutations. This is where the method moves from a two-order

diagnostic to a scalable scheduling procedure.

### 4.5. Compute, Optimizer Scope, and Control

The measured planner time is slightly slower than brute-force evaluation at $k=1$, where both orders are themselves tiny, but it is already faster by $k=5$ and about $13\times$ faster than brute-force pilots at $k=20$ in the Llama and Qwen wall-clock runs (Appendix E.7). This addresses the cost of the required HVPs: the second-order information is not free, but it is cheaper than trying both orders at practical horizons.

AdamW is a stateful optimizer, while the clean derivation applies to memoryless gradient-style updates. We therefore report AdamW as a measured optimizer-scope evaluation and as evidence for the augmented-state direction rather than as the central result: the unmodified SGD-derived predictor reaches 57.1% at AdamW $k=20$ over 1,020 trials with 33.2% regret reduction, and an exploratory Adam-aware augmented-state score reaches 59.6% at $k=10$ with 29.8% regret reduction. AdamW suggests that the correct geometric object is an augmented-state commutator on $(\theta, m, v)$, but developing that theory remains future work. Appendix E.8 gives a local bracket-control ablation. After the shared-drift update, moving a small distance in the predicted target-loss-decreasing bracket direction improves over the uncorrected shared-drift point in 179/204 cases (87.7%) and beats both sequential endpoints in 95/204 cases (46.6%). This is included as a local sanity check on the sign of the bracket direction; the deployed method in this paper remains order selection and tournament ranking.

The same target-domain formulation makes capability preservation an instance of the same planner. If $P$ is a protected capability represented by a loss, set $E = P$: the sign predicts which order gives lower protected loss and the stakes score estimates how much protected performance is at risk. Rehearsal or safety data can also be included as a tournament source when available; no new estimator is required.

## 5. Related Work

**Training order, curricula, and sequential transfer.** Curriculum learning established that training order can affect optimization and generalization (Bengio et al., 2009), and multi-domain or multilingual systems often depend on when domains are emphasized (Choi et al., 2023). Sequential transfer and gradual domain adaptation study related scheduling questions, often through paths or interpolations between distributions (Ruder et al., 2019; Kumar et al., 2020; Zhuang et al., 2024). Closest to our geometric view, Rukhovich et al. (2025) project the Lie bracket of two domains' gradient fields onto a target-loss gradient—the same

bracket-projection construction as our $\sigma = \langle g_E, b \rangle$—but use it as a local optimality criterion that, as they note, is descriptive rather than an ordering algorithm. We instead make that primitive a deployable, held-out *predictor*: we evaluate the target gradient at the shared-drift Trotter reference rather than at $\theta_0$, capturing the $O(\eta^3)$ correction a single-point criterion misses, and add stakes/confidence estimates, subspace HVP computation, and $\eta$-autopilot calibration, validated across four model families, instruction-SFT, DPO, and a diffusion UNet rather than a single bilingual-pretraining setting. Most distinctively, a shared $\theta_0$ reference turns the *same* bracket primitive—not a per-pair drift-matched score—into an $N$-independent weighted tournament whose Borda ranking costs one Hessian-vector product per source, scheduling curricula over dozens of domains.

**Gradient diagnostics, multi-task optimization, and HVPs.** Multi-task methods combine or modify gradients within each update, including adaptive loss reweighting (Chen et al., 2018), game-theoretic formulations (Navon et al., 2022), gradient surgery (Yu et al., 2020), and multilingual variants (Wang et al., 2021; Lee et al., 2022). Gradient-similarity metrics have also been used for task conflicts, forgetting, curricula, and task-order selection (Igarashi et al., 2022; Nguyen et al., 2025; Imanov, 2026). Our cosine baseline confirms that first-order alignment is informative but insufficient: on Llama-3.2-1B it reaches 77.5% accuracy, leaving a 16.2pp gap to the Lie-bracket score (Appendix D.3). The additional cost is Hessian-vector products, which are computable without forming the Hessian (Pearlmutter, 1994) and connect this work to influence-function and geometric numerical perspectives (Koh & Liang, 2017; Pruthi et al., 2020; Hairer et al., 2006). Complementary to our domain-ordering problem, Sweeney (2026) studies update geometry at vocabulary scale through Fisher alignment of shared-output model families; our analysis instead operates at the level of domain interactions and asks how non-commuting training operators compose across sequential curricula.

## 6. Conclusion

We presented a geometric approach to sequential learning based on Lie-bracket commutators of gradient update fields. A directional score projects $H_B g_A - H_A g_B$ onto the target gradient to choose between two domains and attach stakes/confidence estimates. For many domains, a shared-reference variant turns the same bracket primitive into a Lie-Bracket Tournament with one HVP per source and no enumeration of $N!$ schedules. Across pretraining-domain adaptation, instruction-SFT, DPO, and diffusion adaptation, this primitive predicts pairwise order effects, composes into strong sequence-level rankings, and is faster than brute-force pilots at practical horizons.

## Impact Statement

Sequential fine-tuning is ubiquitous, yet the default practice of testing multiple orders doubles compute costs, with associated energy consumption and carbon footprint. Our planner enables principled ordering decisions without running both orders, reducing unnecessary GPU hours. This is particularly valuable in resource-constrained settings where exhaustive ablations on large models are infeasible; confidence gating further indicates when predictions can be trusted directly versus when validation is warranted.

More broadly, this work takes a step toward a geometric theory of curriculum learning. The Lie-bracket lens reveals that training-order effects are not arbitrary but governed by structured second-order interactions. The tournament extension suggests a practical path from pairwise geometry to many-domain scheduling, while the optimizer-scope evaluations clarify when additional validation is prudent.

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

# A. Proofs

## A.1. Proof of Lemma 2.1

*Proof.* Write the update maps as $U_A(\theta) = \theta - \eta g_A(\theta)$ and $U_B(\theta) = \theta - \eta g_B(\theta)$. Then

$$
\begin{aligned}
\theta_{AB} &= U_B(U_A(\theta_0)) \\
&= \theta_0 - \eta g_A(\theta_0) - \eta g_B(\theta_0 - \eta g_A(\theta_0)), \\
\theta_{BA} &= U_A(U_B(\theta_0)) \\
&= \theta_0 - \eta g_B(\theta_0) - \eta g_A(\theta_0 - \eta g_B(\theta_0)).
\end{aligned}
$$

Using first-order Taylor expansion of $g_B$ around $\theta_0$ with integral remainder,

$$
g_B(\theta_0 - \eta g_A(\theta_0)) = g_B(\theta_0) - \eta H_B(\theta_0)\, g_A(\theta_0) + R_B,
$$

where

$$
\begin{aligned}
R_B = \int_0^1 &\Big( H_B(\theta_0 - t\eta g_A(\theta_0)) - H_B(\theta_0) \Big) \\
&\cdot (-\eta g_A(\theta_0))\, dt.
\end{aligned}
$$

By the Lipschitz Hessian assumption, $\|H_B(\theta_0 - t\eta g_A) - H_B(\theta_0)\| \leq M_3\, t\eta \|g_A(\theta_0)\|$, so

$$
\begin{aligned}
\|R_B\| &\leq \int_0^1 M_3\, t\eta \|g_A(\theta_0)\| \cdot \eta \|g_A(\theta_0)\|\, dt \\
&= \frac{M_3}{2}\, \eta^2\, \|g_A(\theta_0)\|^2.
\end{aligned}
$$

Multiplying by the outer $\eta$ that appears in $\theta_{AB}$ yields an $O(\eta^3)$ contribution.

Analogously,

$$
g_A(\theta_0 - \eta g_B(\theta_0)) = g_A(\theta_0) - \eta H_A(\theta_0)\, g_B(\theta_0) + R_A,
$$

$$
\|R_A\| \leq \frac{M_3}{2}\, \eta^2\, \|g_B(\theta_0)\|^2.
$$

Substituting into $\theta_{AB}$ and $\theta_{BA}$ gives

$$
\begin{aligned}
\theta_{AB} &= \theta_0 - \eta(g_A(\theta_0) + g_B(\theta_0)) + \eta^2 H_B(\theta_0)g_A(\theta_0) - \eta R_B, \\
\theta_{BA} &= \theta_0 - \eta(g_A(\theta_0) + g_B(\theta_0)) + \eta^2 H_A(\theta_0)g_B(\theta_0) - \eta R_A,
\end{aligned}
$$

and hence

$$
\begin{aligned}
\theta_{AB} - \theta_{BA} &= \eta^2 \big( H_B(\theta_0)g_A(\theta_0) - H_A(\theta_0)g_B(\theta_0) \big) \\
&\quad + \eta(R_A - R_B).
\end{aligned}
$$

Using the bounds on $R_A, R_B$ yields $\|\eta(R_A - R_B)\| \leq C\eta^3$ with $C = \frac{M_3}{2}(\|g_A(\theta_0)\|^2 + \|g_B(\theta_0)\|^2)$. $\qquad \square$

## A.2. Proof of Proposition 2.3

*Proof.* Let $d := \theta_{AB} - \theta_{BA}$. By the mean value theorem applied to the function $t \mapsto L_E(\theta_{BA} + td)$, there exists $\theta^\star = \theta_{BA} + t^\star d$ for some $t^\star \in (0,1)$ such that

$$
\Delta_E(A, B) = L_E(\theta_{BA} + d) - L_E(\theta_{BA}) = \langle g_E(\theta^\star), d \rangle.
$$

Lemma 2.1 gives $d = \eta^2 b_{AB}(\theta_0) + r_{AB}$ with $\|r_{AB}\| \leq C\eta^3$. Also, $\theta_{BA} - \theta_0 = -\eta(g_A(\theta_0) + g_B(\theta_0)) + O(\eta^2)$, so $\|\theta_{BA} - \theta_0\| = O(\eta)$. Since $d = O(\eta^2)$, we have $\theta^\star = \theta_0 + O(\eta)$. Under locally bounded $H_E$, $g_E$ is locally Lipschitz, hence $g_E(\theta^\star) = g_E(\theta_0) + O(\eta)$. Therefore,

$$
\begin{aligned}
\Delta_E(A, B) &= \langle g_E(\theta_0), \eta^2 b_{AB}(\theta_0) \rangle + O(\eta) \cdot O(\eta^2) + O(\eta^3) \\
&= \eta^2 \langle g_E(\theta_0), b_{AB}(\theta_0) \rangle + O(\eta^3),
\end{aligned}
$$

which is Eq. (10). $\qquad \square$

*Table 5.* Robustness to domain sample size. We split triples by whether they involve low-sample domains and report sign accuracy for the deployable trap_local estimator.

| Subset | #triples | Acc. (trap_local) | $\mathbb{E} \; \lvert\Delta_E\rvert$ |
|---|---|---|---|
| All triples | 1020 | 84.3% | 6.64e-03 |
| No low-data domains | 662 | 85.3% | 7.93e-03 |
| Involves low-data domain | 358 | 82.4% | 4.24e-03 |

### A.3. Proof of Lemma 2.4

*Proof.* The expansions for $\theta_{AB}$ and $\theta_{BA}$ are derived in the proof of Lemma 2.1 (Appendix A.1). Substituting $\theta_{\text{ref}} = \theta_0 - \eta\big(g_A(\theta_0) + g_B(\theta_0)\big)$ yields the stated forms, and the $O(\eta^2)$ closeness follows since the leading correction terms are $O(\eta^2)$ and the remainder is $O(\eta^3)$. □

### A.4. Proof of Proposition 2.5

*Proof.* Let $d := \theta_{AB} - \theta_{BA}$. By the mean value theorem applied to $t \mapsto L_E(\theta_{BA} + td)$, there exists $\theta^\star \in [\theta_{BA}, \theta_{AB}]$ such that $\Delta_E(A, B) = \langle g_E(\theta^\star), d\rangle$. Lemma 2.4 implies $\theta_{AB} = \theta_{\text{ref}} + O(\eta^2)$ and $\theta_{BA} = \theta_{\text{ref}} + O(\eta^2)$, hence $\theta^\star = \theta_{\text{ref}} + O(\eta^2)$. Under $\|H_E(\theta)\| \leq M_{2,E}$, $g_E$ is $M_{2,E}$-Lipschitz, so $\|g_E(\theta^\star) - g_E(\theta_{\text{ref}})\| \leq M_{2,E}\|\theta^\star - \theta_{\text{ref}}\| = O(\eta^2)$. Multiplying by $\|d\| = O(\eta^2)$ (Lemma 2.1) gives $\big|\Delta_E(A, B) - \langle g_E(\theta_{\text{ref}}), d\rangle\big| = O(\eta^4)$. □

## B. Oracle Endpoint Integration Baselines

For completeness, we evaluate *oracle* estimators that require computing both endpoints $\theta_{AB}$ and $\theta_{BA}$ (and therefore are not deployable for order selection). These baselines provide an upper bound on what can be achieved when one is allowed to probe both training orders.

Define the endpoint gradients $g_E(\theta_{AB})$ and $g_E(\theta_{BA})$.

**Oracle trapezoid.** We define

$$\hat{\sigma}_{AB}^{(E)}(\text{trap\_oracle}) := \tfrac{1}{2}\langle g_E(\theta_{AB}) + g_E(\theta_{BA}), b_{AB}(\theta_0)\rangle. \tag{26}$$

**Oracle Simpson.** Let $\theta_{\text{mid}} := \tfrac{1}{2}(\theta_{AB} + \theta_{BA})$ (linear interpolation in parameter space) and compute $g_E(\theta_{\text{mid}})$. Then

$$\hat{\sigma}_{AB}^{(E)}(\text{simpson\_oracle}) := \tfrac{1}{6}\langle g_E(\theta_{AB}) + 4g_E(\theta_{\text{mid}}) + g_E(\theta_{BA}), b_{AB}(\theta_0)\rangle. \tag{27}$$

**Remark (what Simpson does and does not show).** Simpson's rule is a higher-order quadrature approximation for averaging $g_E$ along the same interpolation path, but it requires an additional gradient evaluation at $\theta_{\text{mid}}$ and, crucially, access to both endpoints. Empirically, simpson_oracle is nearly indistinguishable from trap_oracle (see experiments), suggesting that quadrature error along the endpoint segment is not the dominant source of approximation error in our setting.

## C. Robustness to Domain Sample Size

To verify that our results are not driven by low-sample domains, we partition triples into those involving low-data domains (Books3: 61 available samples, YoutubeSubtitles: 77 available samples, out of 80 requested) and those using only high-data domains.

Triples involving low-data domains may exhibit higher variance in gradient estimates due to smaller minibatch coverage, but the sign accuracy remains comparable, confirming that our method is robust to domain sample size variation.

## D. Additional Ablations

**Base (first-order) baseline.** We compute the base (first-order) estimator alongside the main Trotter and trap_local predictors for all runs. For readability, we omit it from Table 2; the main first-order comparator reported in the tables is the gradient-cosine baseline.

*Table 6.* Complete cross-model evaluation results (extends Table 2). Trap_local is a local trapezoid ablation. Regret is mean excess target loss relative to the better order (scaled by $10^4$). Mean effect is the average absolute loss difference $|\Delta_E|$ between orders. LLMs use a final-layer subspace (o_proj, down_proj); DDPM uses convolutional layers.

| Model | Acc. (trotter) | Acc. (trap_local) | Acc. (cosine) | Acc. @25% | Regret (1e4) | Mean effect | $\eta$ |
|---|---|---|---|---|---|---|---|
| Qwen2.5-1.5B | 87.7%±1.8 | 87.1%±2.3 | 61.8% | 95.7%±1.5 | 1.75 | 3.59e-03 | 0.00203 |
| Llama-3.2-1B | 92.0%±2.2 | 92.1%±2.5 | 77.9% | 82.4%±7.9 | 0.72 | 3.82e-04 | 0.00105 |
| Llama-3.1-8B | 82.4%±2.4 | 81.9%±2.3 | 67.2% | 94.9%±1.6 | 2.14 | 5.67e-03 | 0.00133 |
| SmolLM3-3B | 87.2%±2.6 | 78.5%±2.3 | 61.3% | 86.3%±2.5 | 127.8 | 1.25e-01 | 0.00178 |
| DDPM UNet | 91.1%±3.7 | 88.3%±3.7 | 88.9% | 100.0%±0.0 | 0.0000 | 1.99e-07 | 0.09064 |

## D.1. $\eta$-Autopilot

We select a single step size $\eta$ *once per model* from a small pilot run and reuse it for all experiments. The autopilot balances two requirements: (i) *detectability* (order effects must rise above an empirical noise floor), and (ii) *sign validity* (higher-order terms must not destabilize $\text{sign}(\hat{\sigma})$). In practice we run the pilot procedure with three random seeds and take the median selected $\eta$; a single seed usually gives a similar value, but the median is cheap insurance against outliers.

**Detectability floor.** For pilot triples $i$, we compute a deployable commutator score $\hat{\sigma}_i$ and estimate an effective standard error $\sigma_{\text{eff}}$ for the paired loss difference $\Delta_E$ at a small reference step $\eta_{\text{ref}}$ via repeated AB/BA evaluations. Concretely, for each triple we evaluate the per-batch differences $d_{i,j} := L_E(\theta_{AB}; \mathcal{B}_j) - L_E(\theta_{BA}; \mathcal{B}_j)$ over evaluation batches $j = 1, \ldots, m$ and form $\text{SE}_{\Delta,i} := \text{std}_j(d_{i,j})/\sqrt{m}$. We aggregate across pilot triples robustly and use $\sigma_{\text{eff}} = \max\{\text{SE}_\Delta, \sigma_{\text{fp}}\}$, where $\sigma_{\text{fp}}$ is a floating-point floor. An empirical low-signal null floor is logged as a diagnostic but is excluded from the released selection rule, since including it can inflate the detectability floor. Let $S_q := Q_q(|\hat{\sigma}_i|)$ be a high quantile over pilot triples. The score-based detectability constraint $\eta^2 S_q \geq z \, \sigma_{\text{eff}}$ yields Eq. (25). The implementation also computes an analogous loss-difference floor calibrated from the measured $\Delta_E(\eta_{\text{ref}})$; the implemented $\eta_{\text{min}}$ is the conservative maximum of the score-based floor and this $\Delta_{\text{ref}}$ floor, followed by the entanglement correction below. Thus $\eta_{\text{ref}}$ enters the released selection through the $\Delta_{\text{ref}}$ floor even though it does not appear in the algebraic score floor in Eq. (25).

**Sign-validity ceiling.** We compute conservative sign-validity ceilings from two sources: a directional BCH consistency check and a loss-level sign-stability bound for the reference-point score. Let $w_i$ denote nonnegative aggregation weights proportional to the pilot signal magnitude, $w_i \propto |\hat{\sigma}_i|$. The formulas below give the local derivation. The released policy aggregates the directional and loss-level ceilings conservatively and then clamps the selected $\eta$ by the resulting $\eta_{\text{sign}}$; this is the sign-validity ceiling used by the selection rule.

**Tradeoff ceiling.** Independently, we aggregate the BCH magnitude ceilings $\eta_{\text{bch},i}$ to obtain $\eta_{\text{bch}} := Q_{q_\eta}^w(\{\eta_{\text{bch},i}\})$ and define a tradeoff ceiling $\eta_{\text{tradeoff}} := h_{\text{bch}} \eta_{\text{bch}}$, where $h_{\text{bch}} > 1$ allows modest headroom beyond strict magnitude validity (we use $h_{\text{bch}} = 1.5$).

**Per-triple ceiling definitions (sketch).** Let $b_i$ be the bracket vector for triple $i$, and let $d\theta_i(\eta)$ denote the two-step commutator displacement $\theta_{AB} - \theta_{BA}$ computed from two gradient steps at step size $\eta$ using the pilot minibatches. At a small reference step size $\eta_{\text{ref}}$, define the BCH relative error

$$r_{\text{bch},i} := \frac{\|d\theta_i(\eta_{\text{ref}}) - \eta_{\text{ref}}^2 b_i\|}{\|\eta_{\text{ref}}^2 b_i\| + \varepsilon}.$$

Using the $O(\eta^3)$ remainder (so $r_{\text{bch},i}$ scales approximately linearly in $\eta$), we set a magnitude-valid ceiling

$$\eta_{\text{bch},i} := \kappa_{\text{bch}} \frac{\eta_{\text{ref}}}{\max\{r_{\text{bch},i}, \varepsilon\}}.$$

For sign prediction we also compute a directional ceiling using the projection onto the target gradient $g_E$:

$$r_{\text{dir},i} := \frac{|\langle g_E, d\theta_i(\eta_{\text{ref}}) - \eta_{\text{ref}}^2 b_i\rangle|}{|\langle g_E, \eta_{\text{ref}}^2 b_i\rangle| + \varepsilon}, \qquad \eta_{\text{sign},i} := \kappa_{\text{sign}} \frac{\eta_{\text{ref}}}{\max\{r_{\text{dir},i}, \varepsilon\}}.$$

Finally, a loss-level Taylor expansion yields an $\eta^3$ cross-term that can (when oppositely signed) flip $\text{sign}(\Delta_E)$ even if $d\theta$ remains BCH-consistent. Concretely,

$$\Delta_{E,i}(\eta) = \eta^2 \sigma_i + \eta^3 u_i + O(\eta^4), \qquad \sigma_i := \langle g_E, b_i \rangle, \qquad u_i \approx -\langle g_A + g_B, H_E b_i \rangle,$$

so when $\sigma_i u_i < 0$ a first sign flip occurs at $\eta_{\text{flip}} \approx |\sigma_i/u_i|$. We compute a conservative ceiling $\eta_{\text{loss},i}$ from this signed cubic flip condition. In the implementation, the directional and loss-level ceilings are aggregated and guarded separately, then combined with the BCH-headroom and absolute ceilings in the final clamp. The simplified expression $\min\{\eta_{\text{sign},i}, \eta_{\text{loss},i}\}$ is the per-triple local intuition; the released code is the authoritative specification for engineering guards.

**Selection rule (cube-root interpolation with regime gating).** Let

$$\eta_{\text{cube}} := \max\{\eta_{\min}, \eta_{\min}^{2/3} \eta_{\text{tradeoff}}^{1/3}\}, \qquad \alpha_{\min} := (\eta_{\min}/\eta_{\text{sign}})^2,$$

and let $\rho_{\max}$ be a guardrail on how far the tradeoff ceiling can exceed the sign ceiling. The LLM results in Table 2 use the released slack-ratio policy:

$$\eta = \begin{cases} \eta_{\text{sign}}, & \eta_{\min} > \eta_{\text{sign}} \quad \text{(underpowered)} \\ \min\{\eta_{\text{sign}}, \eta_{\text{cube}}\}, & \hat{N}_{\text{eff}} > 1 \quad \text{(entanglement active)} \\ \eta_{\min}, & \alpha_{\min} < \alpha_{\text{strong}} \quad \text{(strongly powered, no entanglement)} \\ \min\{\eta_{\text{sign}}, \eta_{\text{cube}}\}, & \eta_{\text{tradeoff}} \leq \rho_{\max} \eta_{\text{sign}} \\ \eta_{\text{sign}}, & \text{otherwise (sign-limited).} \end{cases}$$

We use $\alpha_{\text{strong}} = 0.1$ and $\rho_{\max} = 2.0$ in our experiments. In all four LLM pilots the N-finder correction is active ($\hat{N}_{\text{eff}} > 1$), so the strongly powered Llama rows in Table 2 use the cube-root-interpolated branch rather than the bare $\eta_{\min}$ branch. The DDPM diffusion model uses the *upper* policy, directly selecting $\eta_{\text{upper}} = \min\{\eta_{\text{sign}}, \eta_{\text{tradeoff}}\}$, because diffusion models' weaker effect magnitudes require less conservative $\eta$ selection to achieve adequate signal-to-noise. For architectures where regime-gating underperforms, forcing cube-root interpolation ($\eta = \eta_{\min}^{2/3} \eta_{\text{tradeoff}}^{1/3}$, clamped to $\eta_{\text{sign}}$) is a robust fallback.

**Entanglement correction (N-finder).** When pilot triples systematically underestimate evaluation-scale effects, we estimate a scale factor $\hat{N}$ by comparing order-effect magnitudes at the same small reference step $\eta_{\text{ref}}$. Let $\Delta_{\text{ref}}$ denote the measured paired loss difference $L_E(\theta_{AB}) - L_E(\theta_{BA})$ at $\eta_{\text{ref}}$ on pilot minibatches. We form (i) a *baseline* set by sampling within-domain triples (all three minibatch draws from a single pilot partition), and (ii) an *entangled probe* set by selecting $(A, B)$ pairs with high gradient cosine overlap and evaluating a small set of high-coupling triples (including an ablation where $E$ is a 50/50 mixture of $A$- and $B$-batches). Let $\delta_{\text{base}} := \text{median}(|\Delta_{\text{ref}}|)$ over baseline triples and $\delta_{\text{probe}} := \text{median}$ of the top-$k$ (largest) $|\Delta_{\text{ref}}|$ values over probe triples. We set $\hat{N} := \max\{1, \delta_{\text{probe}}/\delta_{\text{base}}\}$ and rescale $\eta_{\min} \leftarrow \eta_{\min}/\sqrt{\hat{N}_{\text{eff}}}$ before applying the selection rule above, using a conservative $\hat{N}_{\text{eff}} \geq 1$ (e.g. a bootstrap lower confidence bound on $\hat{N}$ when available).

## D.2. $\eta$ Sweeps (Robustness Check)

To validate the predicted $\eta^2$ scaling and demonstrate robustness across operating points, we evaluated the method at multiple $\eta$ values per model. Table 7 shows results for $\eta \in \{0.0003, 0.001, 0.003\}$ for LLMs and $\eta \in \{0.0003, 0.001, 0.003, 0.01, 0.03, 0.1\}$ for diffusion. These sweeps confirm that (i) $\mathbb{E}|\Delta_E|$ increases with $\eta$ as predicted by theory (Appendix A.2), and (ii) sign accuracy can vary substantially with $\eta$, motivating principled selection.

**Key observations.** Across models, accuracy is sensitive to the step size, and effect magnitudes grow rapidly with $\eta$. Table 2 therefore uses a single $\eta$ chosen by our autopilot from a small disjoint pilot set, while the sweeps here serve as a robustness check and to illustrate the range of valid operating points. Table 7 summarizes the representative sweep points; we omit the full per-seed grid for space.

## D.3. Gradient Cosine Similarity Baseline

To validate that second-order corrections are necessary, we compare against a natural first-order baseline: selecting the order with higher gradient-target cosine similarity. While gradient similarity has been used to detect task conflicts in multi-task

*Table 7.* Robustness check: Trotter accuracy across $\eta$ sweep (mean ± std over seeds).

| Model | $\eta$ | Trotter Acc | Mean $|\Delta_E|$ | Note |
|---|---|---|---|---|
| Qwen2.5-1.5B | 0.0003 | 56.2%±2.8 | 1.06e-04 | |
| | 0.001 | 70.3%±5.5 | 8.15e-04 | |
| | 0.003 | 84.2%±3.8 | 6.64e-03 | Sweep peak |
| Llama-3.2-1B | 0.0003 | 92.6%±1.0 | 1.44e-05 | Sweep peak |
| | 0.001 | 92.4%±2.0 | 3.13e-04 | |
| | 0.003 | 83.5%±1.8 | 2.38e-03 | |
| Llama-3.1-8B | 0.0003 | 61.8%±0.3 | 4.84e-04 | |
| | 0.001 | 76.0%±2.5 | 3.85e-03 | |
| | 0.003 | 82.4%±2.5 | 1.42e-02 | Sweep peak |
| SmolLM3-3B | 0.0003 | 81.6%±2.8 | 1.81e-03 | |
| | 0.001 | 87.2%±2.1 | 2.61e-02 | Sweep peak |
| | 0.003 | 82.2%±2.2 | 4.83e-01 | |
| DDPM UNet | 0.0003 | 49.1%±2.6 | 5.48e-08 | |
| | 0.001 | 49.1%±8.0 | 6.47e-08 | |
| | 0.003 | 41.7%±6.8 | 6.37e-08 | |
| | 0.01 | 50.9%±15.9 | 6.34e-08 | |
| | 0.03 | 60.2%±7.3 | 6.67e-08 | |
| | 0.10 | 85.2%±4.7 | 2.08e-07 | Sweep peak |

learning (Yu et al., 2020) and measure forgetting in continual learning (Imanov, 2026), to our knowledge it has not been applied as a predictor for sequential fine-tuning order.

**Baseline definition.** Given source domains $A$, $B$ and target domain $E$, compute at the initial point $\theta_0$:

$$\cos(g_A, g_E) = \frac{\langle g_A, g_E \rangle}{\|g_A\|\|g_E\|}, \qquad \cos(g_B, g_E) = \frac{\langle g_B, g_E \rangle}{\|g_B\|\|g_E\|}.$$

Predict $A \to B$ if $\cos(g_B, g_E) > \cos(g_A, g_E)$ (i.e., end at the domain more aligned with $E$), otherwise predict $B \to A$.

**Results on Llama-3.2-1B.** We evaluated this baseline on the full 204-triple test suite for Llama-3.2-1B (float32 precision, $\eta = 0.000936$, 4 evaluation batches). The cosine baseline achieves **77.5%** sign accuracy, substantially better than random (50%) but leaving a **16.2pp gap** that the Lie-bracket method fills (93.6%).

**Interpretation.** This confirms that (i) first-order gradient geometry captures *some* predictive signal—alignment with the target matters—but (ii) second-order Lie-bracket corrections are necessary to achieve strong performance. The 16.2pp gap validates the cost of computing Hessian-vector products: simple gradient cosine similarity, while informative, is insufficient for reliable transfer-order prediction.

### D.4. Multi-Step Generalization

To validate practical relevance beyond the $k=1$ regime, we test whether the predictor (computed *once* at $\theta_0$ using single-step theory) remains accurate when ground truth is evaluated at $k>1$ steps per domain.

**Experimental setup.** For each triple $(A, B, E)$, we:

1. Compute the predictor $\text{sign}(\hat{\sigma})$ at $\theta_0$ using the $k=1$ Lie-bracket theory (standard Trotter scoring)

2. Evaluate ground truth at multiple $k$ values: for each $k \in \{1, 5, 10, 20\}$, take $k$ gradient steps on domain $A$ followed by $k$ steps on domain $B$ (and vice versa), then compare final losses $L_E(\theta_{AB}^{(k)})$ vs. $L_E(\theta_{BA}^{(k)})$

3. Check if the $k=1$ predictor correctly predicts the sign of $\Delta_E^{(k)} := L_E(\theta_{AB}^{(k)}) - L_E(\theta_{BA}^{(k)})$

Importantly, the predictor is held fixed across all $k$: we do *not* recompute $\hat{\sigma}$ after taking any training steps. We test on Llama-3.2-1B and Qwen2.5-1.5B with 204 triples each (single seed). Llama uses $\eta{=}0.000936$ and float32 precision; Qwen uses $\eta{=}0.00203$ and bfloat16 precision (with fp32-cast parameters). To control runtime for multi-step ground truth, we use a fixed evaluation protocol for all compared orderings.

**Results.** The original single-seed short-horizon validation gave Qwen2.5-1.5B accuracies of 91.2%, 84.3%, 77.0%, and 71.6% for $k \in \{1, 5, 10, 20\}$, and Llama-3.2-1B accuracies of 93.6%, 81.9%, 76.0%, and 71.6%. The multi-seed expansion supersedes this smaller table for Llama-3.2-1B; see Table 10 for the 1,020-trial validation through $k{=}50$.

**Interpretation.** The predictor works best in the $k{=}1$ regime where the second-order theory is exact. At longer horizons ($k{\geq}10$), accuracy degrades as higher-order trajectory effects accumulate (the Lie bracket evolves along the optimization path, and our predictor only captures the initial $\theta_0$ geometry). However, the predictor remains substantially above random even at $k{=}20$, suggesting the initial second-order geometry has persistent predictive power.

This validates practical relevance: the method is most reliable for short-to-medium adaptation horizons ($k{\leq}10$ steps per domain, which covers many fine-tuning scenarios), and can inform pilot studies even for longer training runs.

### D.5. Long-Horizon Validation ($k = 50$)

A common concern is that order effects become dominated by optimization noise at long horizons. We therefore run a compute-limited validation at $k{=}50$ steps per domain (100 updates total) on Llama-3.2-1B, holding the $k{=}1$ predictor fixed at $\theta_0$ as in Appendix D.4. In the expanded multi-seed run, the predictor achieves 666/1020 = 65.3% sign accuracy at $k{=}50$ (exact two-sided binomial test: $p = 9.7 \times 10^{-23}$) and reduces mean regret by 54.7% relative to random ordering.

### D.6. Expanded-Subspace (Multi-Layer) Validation

For LLMs, our main results use a final-layer subspace for computational efficiency. Here we test whether the predictor remains effective when the trainable set spans multiple transformer layers, so that curvature computation must backpropagate through subsequent attention blocks. We run Llama-3.2-1B with an 8-layer trainable subset (layers 8–15) using `o_proj+down_proj` in each layer (16 weight matrices; $8\times$ more than final-layer tuning), with $\eta{=}0.001376$ selected by $\eta$-autopilot calibrated for this larger subspace. Across the expanded multi-seed evaluation, the Trotter predictor obtains 825/1020 sign accuracy (80.9%) with 22.3% regret reduction.

### D.7. AdamW Robustness

Our theoretical analysis assumes memoryless gradient descent updates. AdamW is stateful: bias correction, momentum, adaptive second moments, and weight decay enlarge the dynamical state, so a faithful commutator should live in an augmented state space rather than only in parameter space. For that reason, we treat AdamW as a limitation and partial-extension test rather than as a solved setting.

**Unmodified SGD-derived predictor.** We evaluate AdamW ground truth while keeping the predictor fixed at $\theta_0$ and computed from the SGD-style bracket. Over 1,020 Llama-3.2-1B trials, the unmodified predictor is weak at short horizons, but improves modestly by $k{=}20$. Table 8 replaces the earlier optimistic single-seed AdamW numbers with the larger corrected evaluation.

*Table 8.* Corrected AdamW robustness. The unmodified SGD-derived predictor is only a modest partial extension under AdamW; stateful optimizers require augmented-state commutators.

| AdamW horizon | Correct/total | Accuracy | Regret red. |
|---|---|---|---|
| $k{=}5$ | 487/1020 | 47.7% | -9.2% |
| $k{=}10$ | 523/1020 | 51.3% | 12.1% |
| $k{=}20$ | 582/1020 | 57.1% | 33.2% |
| $k{=}50$ | 476/1020 | 46.7% | -2.0% |

**Adam-aware exploratory variant.** As a first step toward the correct optimizer-aware theory, we also tested an Adam-aware augmented-state score that accounts for optimizer state in the local update geometry. At $k{=}10$, this exploratory

variant obtains 365/612 correct predictions (59.6%) with 29.8% regret reduction. We include the result to show a plausible path forward, but we do not use it as a headline claim.

**Interpretation.** The conclusion is not that AdamW is solved. Rather, the SGD-style bracket remains somewhat informative once the adaptive transient is less dominant, while the correct treatment of AdamW should augment the state and derive commutators for the optimizer dynamics. In particular, AdamW suggests that the correct geometric object is an augmented-state commutator on $(\theta, m, v)$, but developing that theory remains future work. This is an important limitation and a natural direction for future work.

### D.8. Finite-Difference HVP Ablation

To test whether approximate curvature information suffices, we evaluated finite-difference HVP approximation:

$$H \cdot v \approx \frac{\nabla L(\theta + \epsilon v) - \nabla L(\theta)}{\epsilon}$$

This avoids 2nd-order autodiff (`create_graph=True`) at the cost of two gradient passes instead of one double-backward.

We tested on Qwen2.5-1.5B with 50 triples (seed 0) at $\eta=0.0026$, comparing exact HVPs (Pearlmutter's trick) against finite-difference approximation with $\epsilon=10^{-5}$. Table 9 shows accuracy by $k$.

*Table 9.* Exact vs finite-difference HVP approximation. FD-HVP degrades accuracy by 4–12pp across all $k$ values.

| Method | $k=1$ | $k=2$ | $k=5$ | $k=10$ | $k=20$ |
|---|---|---|---|---|---|
| Exact HVP | **94%** | **90%** | **84%** | **66%** | **68%** |
| FD-HVP ($\epsilon=10^{-5}$) | 82% | 78% | 76% | 62% | 64% |
| $\Delta$ (pp) | -12 | -12 | -8 | -4 | -4 |

The consistent degradation across $k$ values indicates that the predictor is sensitive to curvature precision. While FD-HVP has comparable computational cost to exact HVPs (both $\sim 2\times$ a gradient), the approximation error degrades accuracy substantially. This validates that precise second-order information is necessary to capture the Lie-bracket geometry governing order effects.

## E. Extended Empirical Evidence

### E.1. Theory-Validation Figures

This appendix preserves the original ablation material above and gives additional evidence for the long-horizon, optimizer, post-training, and many-domain claims in the main text. Unless otherwise stated, large-$N$ tournament percentiles are computed against 500 sampled random curricula, not against all $N!$ possible orders.

### E.2. Full Long-Horizon Table

Table 10 reports the full multi-seed long-horizon result used in the main text. The predictor is held fixed at $\theta_0$, so degradation with $k$ reflects accumulated higher-order trajectory effects rather than predictor recomputation.

*Table 10.* Long-horizon validation over 1,020 trials.

| Horizon | Correct/total | Accuracy | Regret reduction |
|---|---|---|---|
| $k=1$ | 949/1020 | 93.0% | 64.0% |
| $k=5$ | 962/1020 | 94.3% | 97.4% |
| $k=10$ | 912/1020 | 89.4% | 93.1% |
| $k=20$ | 831/1020 | 81.5% | 86.3% |
| $k=50$ | 666/1020 | 65.3% | 54.7% |

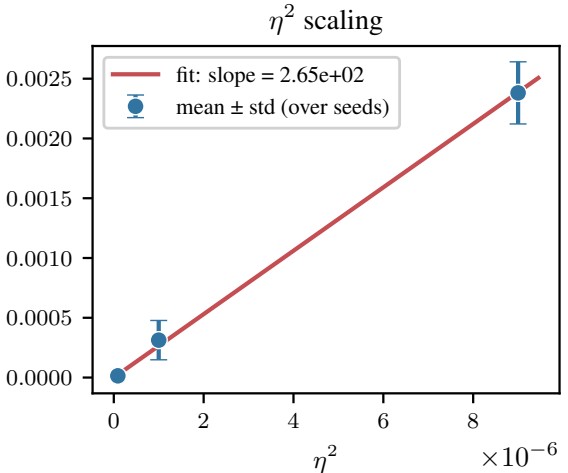

*Figure 1.* $\eta^2$ scaling on Llama-3.2-1B across three step sizes (5 seeds each). The mean absolute loss difference $|\Delta_E|$ scales linearly with $\eta^2$, consistent with the BCH expansion's leading-order term.

*Figure 2.* BCH alignment on Llama-3.2-1B. The cosine similarity $\cos(d\theta, \eta^2 b)$ concentrates near 1 across all step sizes, confirming directional alignment with the Lie-bracket prediction.

### E.3. Multi-Layer Subspace Validation

The original submission validated a final-layer trainable subspace for LLMs. An expanded multi-layer Llama-3.2-1B run tunes the last 8 transformer layers (layers 8–15; o_proj+down_proj in each) and evaluates 1,020 trials. The Trotter predictor obtains 825/1020 sign accuracy (80.9%) with 22.3% regret reduction. We include this result as a robustness check that the signal is not confined to a tiny final-layer subspace. A separate eager-attention ablation on a 50-triple probe obtains the same 92.0% Trotter sign accuracy when using only post-attention layers (o_proj, down_proj) and when adding attention input projections (q_proj, k_proj, v_proj); the post-attention restriction is therefore a systems and memory compromise, not a theoretical exclusion.

### E.4. Sequence-Level $N{=}3$ Scheduling

For triples of source domains, all $3! = 6$ orders can be evaluated exactly. Across 40 quadruples, exact tournament scoring recovers the best sequence in 87.5% of cases and places it in the top two in 96.0% of cases. The scalable Borda variant reaches 79.0% top-1 and 94.5% top-2, with mean Spearman correlation 0.929 and mean regret reduction 93.5% for exact scoring. This experiment is the cleanest direct evidence that pairwise bracket information composes into sequence-level rankings.

### E.5. Large-$N$ Tournament Details

Table 11 reports the large-$N$ sampled-percentile evaluations used for the paper's large-domain claims. Each percentile compares the bracket-derived order against 500 sampled random curricula; the MMLU ranges report the two seeds included in the large-$N$ response. The important point is not an exhaustive optimum over $N!$ schedules, which is infeasible at these sizes, but that the bracket tournament produces high-percentile orders while simple magnitude-only sorting is not a target-conditioned curriculum rule. The MMLU $N{=}56$ rows are the clearest illustration among the reported comparisons: the descending gradient-norm baseline falls below the first sampled percentile even though the bracket tournament reaches the 99.0–99.6th percentile.

Each pairwise tournament edge is an intrinsic local quantity for a pair of domains and a target. With the shared-reference convention used for Borda ranking, the edge between domains $i$ and $j$ is

$$W_{ij} = \langle g_j, H_i g_E^{\text{ref}} \rangle - \langle g_i, H_j g_E^{\text{ref}} \rangle,$$

so adding or removing other source domains changes the set of edges being aggregated but not any existing $W_{ij}$. We verified this directly by recomputing overlapping bracket entries in $N{=}5$, $N{=}7$, and $N{=}10$ runs: all 41 overlapping entries are

*Table 11.* Large-$N$ tournament evaluations used for the paper's large-domain claims. Percentiles are relative to 500 sampled random curricula. "Grad" is descending gradient-norm ordering.

| Dataset/target | Setting | Bracket tournament | Grad | Notes |
|---|---|---|---|---|
| MMLU subjects | $N=56, k=1$ | 99.0–99.6 | $< 1$ | 2 seeds |
| MMLU subjects | $N=30, k=5$ | 89–96 | 2–22 | 2 seeds |
| Stack Python | $N=85, k=1$ | 99 | 69 | 85 available languages |
| Stack Python | $N=10, k=5$ | 86 | 47 | programming-language subset |
| Dolly summarization | $N=7, k=5$ | 99.8 | 49.9 | target-specific SFT task |

bit-for-bit identical at float32 precision, and pairwise sign accuracy remains stable at 80%, 76%, and 73% respectively (the $N=10$ pairwise test has $p = 0.0025$ over 45 pairs). This supports the view that the pairwise signal does not weaken merely because more domains are present; what changes with $N$, horizon length, and target is the sequence-level aggregation problem built on top of those edges.

### E.6. Instruction-SFT and DPO Details

Instruction-SFT uses Dolly-style task categories as domains. At $k=1$, the Trotter predictor obtains 471/480 correct signs (98.1%), mean Spearman 0.940, and 96.2% recovered fraction. At $k=20$, it obtains 351/480 correct signs (73.1%) and 58.0% regret reduction.

For DPO, we evaluate offline preference optimization domains. At $k=1$, the predictor obtains 352/356 correct signs (98.9%) with mean pilot BCH cosine 0.999. At $k=20$, it obtains 257/356 correct signs (72.2%) and 54.7% regret reduction. We describe this as DPO/offline preference optimization rather than online RLHF.

### E.7. Wall-Clock Measurements

Table 12 reports the wall-clock comparisons. The planner is not cheaper than brute force at $k=1$, where evaluating both orders is itself very small; it becomes substantially cheaper at the horizons relevant for pilot fine-tuning.

*Table 12.* Wall-clock measurements in seconds.

| Model | Horizon | Planner | Brute force | Speedup |
|---|---|---|---|---|
| Llama-3.2-1B | $k=1$ | 0.811 | 0.743 | 0.92× |
| Llama-3.2-1B | $k=5$ | 0.811 | 2.853 | 3.52× |
| Llama-3.2-1B | $k=20$ | 0.811 | 10.764 | 13.27× |
| Llama-3.2-1B | $k=50$ | 0.811 | 26.599 | 32.79× |
| Qwen2.5-1.5B | $k=1$ | 0.670 | 0.597 | 0.89× |
| Qwen2.5-1.5B | $k=5$ | 0.670 | 2.290 | 3.42× |
| Qwen2.5-1.5B | $k=20$ | 0.670 | 8.675 | 12.94× |
| Qwen2.5-1.5B | $k=50$ | 0.670 | 21.402 | 31.92× |

### E.8. Bracket-Control Experiment

The main planner selects between sequential endpoints; the same bracket vector also defines a local correction direction around the shared drift point. Let

$$\theta_{\text{ref}} = \theta_0 - \eta(g_A + g_B), \qquad \hat{\sigma} = \langle g_E(\theta_{\text{ref}}), b_{AB} \rangle. \tag{28}$$

A first-order expansion around $\theta_{\text{ref}}$ gives

$$L_E(\theta_{\text{ref}} + \mu\eta^2 b_{AB}) = L_E(\theta_{\text{ref}}) + \mu\eta^2\hat{\sigma} + O(\eta^4), \tag{29}$$

so choosing $\mu = -c\,\text{sign}(\hat{\sigma})$ with $c > 0$ moves locally in the predicted target-loss-decreasing direction. With $c = 0.5$, the resulting correction $\theta_{\text{ctrl}} = \theta_{\text{ref}} - 0.5\,\text{sign}(\hat{\sigma})\eta^2 b_{AB}$ improves over the uncorrected shared-drift update on the Llama-3.2-1B Pile-domain triples in 179/204 cases (87.7%) and beats both sequential endpoints in 95/204 cases (46.6%).

