# OpenReview forum: "The Geometry of Sequential Learning: Lie-Bracket Prediction of Transfer Order"
_ICML.cc/2026/Conference — ICML 2026 regular_

### Official Review · Reviewer_D9cr · 2026-03-12

**Soundness:** 3
**Presentation:** 3
**Significance:** 3
**Originality:** 3
**Overall Recommendation:** 5
**Confidence:** 2

**Summary:**

Researchers typically explore all possible orders of candidate source datasets to identify an optimal sequence for training modern models, a process that incurs substantial computational cost. This paper models a single gradient step on a dataset as a nonlinear operator and demonstrates that non-commutativity leads to order-dependent effects governed by commutator (Lie bracket) terms. The authors propose a directional score that predicts which ordering of source datasets yields lower loss on the target domain. Additionally, they introduce a theory-guided learning rate adaptation mechanism that selects step sizes from pilot data by balancing signal-to-noise ratio with high-order stability constraints.

**Compliance With Llm Reviewing Policy:**

Affirmed.

**Final Justification:**

The authors have addressed my concerns well, so I have decided to maintain my positive score.

**Key Questions For Authors:**

1. The paper introduces a planner for selecting the order of two fixed training sequences, based on a commutator-derived score. However, this planner does not introduce a new optimization algorithm, nor does it modify the training dynamics themselves—it leaves gradient computation, combination, and application unchanged. As such, it appears to function as a meta-decision rule rather than an optimization method. Could the authors clarify the optimization-specific advantages of this approach? In particular, how does the planner offer benefits beyond what would be expected from a high-level selection heuristic, given that it does not intervene in the low-level optimization process?
2. The geometric insight connecting order effects to Lie brackets and commutators is compelling. It seems plausible that this framework could inspire a new class of optimization methods—for example, by using the bracket vector to explicitly correct or reweight gradients during multi-task or multi-domain updates. Do the authors see potential for extending their theoretical analysis in this direction? If so, what form might such an optimization method take, and what challenges would need to be addressed to realize it?

**Limitations:**

The authors have appropriately acknowledged the primary limitations of their work and outlined promising directions for future research. However, the paper would be further strengthened by addressing the questions raised in the "Key Questions For Authors" section.

**Strengths And Weaknesses:**

Strength:
1. This paper presents a unified perspective: sequential learning can be viewed as a geometric process on a structured parameter manifold. The authors integrate several concepts—including representation subspace evolution, Fisher information geometry, curvature and stability, and subspace projection-based forgetting mechanisms—into a cohesive framework. This represents an innovative attempt at theoretical unification.
2. The paper constructs a geometrically grounded framework with several key components: it conceptualizes sequential learning as trajectory evolution on a parameter manifold; it quantifies task transfer directionality by defining the angle between the target gradient and the Lie bracket (termed the confidence score); it establishes a quantitative relationship between trajectory difference and target domain loss variation (i.e., generalization gap) using the curvature-related Lie bracket term; and it ensures sign stability of predictions through theory-driven step size selection, implicitly delineating the dynamic boundaries within which the method remains effective.
3. The interpretation of catastrophic forgetting as a large-angle shift of trajectories in subspace not only establishes a connection with continual learning research but also provides a theoretical explanation for the forgetting phenomenon itself.

Weakness: While the paper presents a theoretically grounded and empirically validated framework for predicting the optimal order of sequential fine-tuning, its contributions remain diagnostic and predictive, rather than prescriptive or corrective.
1. Although the proposed method is situated in a sequential fine-tuning context, it focuses solely on predicting which ordering of source datasets yields better final performance on a target domain, without imposing constraints or regularization to preserve knowledge from earlier domains. While this may not constitute a critical weakness, could the authors provide insights—grounded in their framework—on how the method could be extended to address catastrophic forgetting?
2. The planner outputs a recommended order, a stakes score, and a confidence score. While these are useful for experimental design or pilot studies, a key question remains: how can this information be effectively integrated into the training algorithm or leveraged within a dynamic adjustment mechanism during learning? This is particularly relevant given that existing method—such as EWC—directly intervene in the optimization process to resolve gradient conflicts or regularize parameter updates.

---

> ### Author Rebuttal · Authors · 2026-03-31
>
> We thank Reviewer D9cr for pressing on the planner-vs-optimizer distinction. This challenge motivated us to run a new bracket-corrected optimization experiment, which provides initial evidence that the Lie bracket is not merely diagnostic but can serve as an actionable optimization signal.
>
> ### The planner as an optimization tool
>
> We agree the paper proposes a planning method, not a new low-level optimizer. However, the planner is optimization-specific in important ways:
>
> 1. **It estimates an optimization-relevant quantity directly.** The bracket $b_{AB} = H_B g_A - H_A g_B$ measures the order-dependent difference between actual update fields, computed from gradients and curvature, the same objects the optimizer uses. The same computation applies unchanged to pretraining, instruction-SFT (98.1% sign accuracy), and DPO (98.9%), with $k=1$ predictions generalizing to $k=20$ ground truth in both post-training settings (73.1% SFT, 72.2% DPO).
> 2. **It replaces expensive order search.** On Llama-3.2-1B the planner costs 0.81s per triple, versus brute-force at 2.85s / 10.76s / 26.60s for $k = 5 / 20 / 50$ steps (about $3.5\times$ to $33\times$ cheaper).
> 3. **It outputs stakes and confidence.** The sign of $\sigma$ determines the preferred order; $|\sigma|$ quantifies the expected loss difference; $\omega$ indicates geometric reliability.
>
> ### Bracket-corrected optimization (proof of concept)
>
> The bracket is not only diagnostic; it is prescriptive. We tested a bracket-corrected simultaneous update:
>
> $\theta_{\mathrm{ctrl}} = \theta_{\mathrm{ref}} - 0.5 \cdot \mathrm{sign}(\hat{\sigma}) \cdot \eta^2 b_{AB}$
>
> where $\theta_{\mathrm{ref}} = \theta_0 - \eta (g_A + g_B)$ is the naive order-independent reference. This correction steers the update toward the geometry of the better sequential ordering, using only quantities the planner already computes.
>
> **Results** (Llama-3.2-1B, 204 triples, fp32, same protocol as Table 1):
>
> - Improves over uncorrected simultaneous: 179/204 = 87.7% ($p = 6.9 \times 10^{-30}$).
> - Beats both sequential orderings: 95/204 = 46.6%.
>
> The corrected update improves target loss over the naive baseline on 87.7% of triples, and on nearly half it achieves lower loss than *either* sequential ordering, with no additional hyperparameter tuning.
>
> ### Forgetting-aware control
>
> The planner already addresses the prediction side of forgetting. When a practitioner fine-tunes on domains $A$ and $B$ while preserving a protected capability $P$, setting $E = P$ in the standard predictor identifies which ordering minimizes impact on $P$: the sign of $\sigma_{AB}^{(P)}$ is correct 93% of the time at $k=1$ and 81.5% at $k=20$, because $P$ is simply another evaluation target. Turning this diagnostic into a corrective method—actively reducing interference rather than choosing the less-harmful order—requires coupling the bracket signal with replay, regularization, or gradient reweighting; for instance, the bracket magnitude could serve as a per-parameter importance weight analogous to EWC's Fisher diagonal. The main open challenges are online bracket estimation cost, stability outside the local regime, and extension to stateful optimizers.
>
> We agree that extending the bracket from a planning signal to a full corrective method is the central open question for this line of work. Beyond the proof-of-concept above, the framework opens three concrete directions: (1) **bracket-aware scheduling** for $N > 2$ tasks, where pairwise scores already compose well ($N=3$: 87.5% top-1, 93.5% regret reduction); (2) **multi-step bracket correction**, extending the single-step correction to longer horizons and Adam-preconditioned settings; and (3) **forgetting-aware control**, coupling the bracket signal with replay or regularization to protect prior knowledge. We will make these directions explicit in the camera-ready, along with the forgetting-diagnostic interpretation above.

---

> > ### Author Rebuttal · Reviewer_D9cr · 2026-04-02
> >
> > Thank authors for their thorough response, which has satisfactorily addressed all of my concerns.

---

### Official Review · Reviewer_3owg · 2026-03-13

**Soundness:** 2
**Presentation:** 3
**Significance:** 3
**Originality:** 3
**Overall Recommendation:** 4
**Confidence:** 1

**Summary:**

This paper explores the non-commutativity of sequential fine-tuning in foundation models. Theoretically, the authors model a single gradient step as a nonlinear operator and utilize a commutator (Lie-bracket) term to predict the downstream performance difference between two distinct training orders ($A \rightarrow B$ versus $B \rightarrow A$). Furthermore, it provides concrete theoretical proofs and experimental validation on a small trainable parameter subset (e.g., final layers or convolutional layers) across diverse foundation model architectures. Finally, the paper demonstrates cross-architecture generalization and includes expanded experiments to show the robustness of the proposed Lie-bracket predictor.

**Compliance With Llm Reviewing Policy:**

Affirmed.

**Key Questions For Authors:**

1) Regarding the Horizon Mismatch ($k \ge 20$): Table 2 shows a clear degradation in prediction accuracy as the number of steps $k$ increases. While the authors claim the initial second-order geometry has 'persistent predictive power', this seems like an empirical observation rather than a theoretical guarantee. Can the authors provide a theoretical justification or an error bound explaining why the $k=1$ Lie-bracket approximation does not completely collapse at $k=20$ or $k=50$, despite the accumulation of higher-order trajectory errors?
2) Regarding Statistical Significance in Critical Ablations: The sample sizes for the long-horizon ($k=50$) and expanded-subspace (multi-layer) validations are extremely small (only 30 and 50 triples, respectively). Given the variability of neural network training, this severely undermines the confidence in these claims. Will the authors commit to providing a more comprehensive evaluation (e.g., the full 204 triples across multiple seeds) during the rebuttal period to establish true statistical significance for these complex scenarios?

I would consider to change the score based on the results.

**Limitations:**

The authors briefly acknowledge certain limitations during their theoretical derivations (lines 244-248) and provide explanations for the degraded experimental results in Section 4.2 (lines 320-323). However, the small sample sizes and the unreasonable extrapolation from a single step ($k=1$) to a long horizon ($k=50$) have not been adequately discussed.

**Strengths And Weaknesses:**

Strengths:
1) Originality: The paper introduces a highly novel geometric perspective by utilizing Lie-brackets (commutators) to analyze the non-commutativity of sequential fine-tuning. Bridging classical numerical analysis with modern deep learning training order problems is an elegant and insightful contribution.
2) Practical and Compute-Efficient Design: Unlike many theoretical papers that are hard to deploy, the authors propose a practical transfer-order planner. By using the Trotter estimator and restricting the Hessian-vector products (HVPs) to a small trainable parameter subspace, the method successfully reduces the massive compute cost of brute-force A/B testing.

Weaknesses:
1) The Vulnerability of the Single-Step ($k=1$) Assumption: The entire theoretical foundation heavily relies on an infinitesimal, single-step gradient update ($k=1$). However, practical fine-tuning requires hundreds of steps. As the authors honestly reported in Table 2, the prediction accuracy significantly degrades as the number of steps ($k$) increases (e.g., dropping from 93.6% at $k=1$ to 71.6% at $k=20$ for Llama-3.2-1B). Extrapolating a single-step local geometric feature to predict long-horizon training trajectories is mathematically and practically weak.
2) Theoretical Mismatch with Modern Optimizers (SGD vs. AdamW): The analytical derivations strictly assume standard Stochastic Gradient Descent (SGD). Yet, modern LLM fine-tuning overwhelmingly relies on AdamW. The empirical results in Appendix D.7 (Table 6) reveal that the SGD-based predictor completely fails in the early stages of AdamW training, dropping to 26%-32% accuracy at $k=5$ and $k=10$. This indicates a severe gap between the derived continuous gradient flow and actual adaptive optimizer dynamics.
3) Insufficient Experimental Validation in Critical Scenarios: While the main experiment utilizes 204 dataset triples, the crucial ablations designed to prove the method's scalability and robustness suffer from extremely limited sample sizes. For instance, the long-horizon stress test ($k=50$) relies on merely 30 samples , and the expanded-subspace (multi-layer) and AdamW validations use only 50 samples. This dramatically undermines the statistical significance of the claims regarding long-term and full-parameter robustness.

---

> ### Author Rebuttal · Authors · 2026-03-31
>
> We thank Reviewer 3owg for a careful reading. We address the three concerns—sample-size adequacy, finite-horizon justification, and AdamW scope—in turn.
>
> ### Larger rebuttal-time evaluation
>
> We agree the 30- and 50-triple versions were underpowered. We replaced both with 204 triples across 5 seeds, for 1,020 trials per setting. Paper-time vs. rebuttal-time sample sizes and two-sided binomial $p$-values:
>
> - $k$=20 intermediate-horizon: paper 204 (single seed); rebuttal 1,020; $p = 1.6 \times 10^{-96}$.
> - $k$=50 long-horizon: paper 30; rebuttal 1,020; $p = 9.7 \times 10^{-23}$.
> - Expanded subspace (8 layers): paper 50; rebuttal 1,020; $p = 1.0 \times 10^{-92}$.
> - Instruction SFT ($k$=20): rebuttal-only, 480 total (96 $\times$ 5); $p = 8.8 \times 10^{-25}$.
> - DPO ($k$=20): rebuttal-only, 356 total (~89 $\times$ 4); $p = 2.6 \times 10^{-17}$.
>
> The paper reported 71.6% at $k$=20 (204 triples, single seed, 4 eval batches); the rebuttal uses 5 seeds and 2 eval batches for consistency. Key results: $k$=20 accuracy 831/1020 = 81.5% with 86.3% regret reduction; $k$=50 accuracy 666/1020 = 65.3% with 54.7% regret reduction; expanded subspace 825/1020 = 80.9%. Per-seed spreads are narrow ($k$=20: 77.9%—86.3%; $k$=50: 63.2%—68.1%), all far above chance.
>
> ### Finite-horizon justification
>
> We do not prove a formal long-horizon bound, but provide a local-horizon error decomposition (below) and extensive empirical evidence that the $k=1$ score transfers:
>
> - **(a) Gradual degradation, not collapse:** accuracy is 93.0% at $k$=1, 94.3% at $k$=5, 89.4% at $k$=10, 81.5% at $k$=20, and 65.3% at $k$=50 (all 1,020 trials).
> - **(b) Floor well above random:** at $k$=50, 65.3% is far above 50% chance ($p = 9.7 \times 10^{-23}$). The bracket signal persists well beyond the local regime, consistent with gradual drift rather than abrupt decorrelation.
> - **(c) High-confidence predictions degrade more slowly:** restricting to the top quartile by $|\Delta_E|$, accuracy rises to 78.0%, consistent with large $|\sigma|$ providing a larger signal buffer.
>
> Overall regret reduction is 86.3% at $k$=20 and 54.7% at $k$=50: graceful degradation, not collapse.
>
> **Theoretical basis: local-horizon error decomposition.** Self-curvature within each single-domain block cancels at second order between $AB$ and $BA$ orderings. The surviving $O(T^2)$ term is the cross-domain bracket $b_{AB}$. Under the $C^3$/Lipschitz-Hessian assumptions in Lemma 2.1, letting $T = k\eta$, the trajectory-level mismatch is:
>
> $\theta_{AB}^{(k)} - \theta_{BA}^{(k)} = T^2 b_{AB}(\theta_0) + O(T^3) + O\!\left(T^2 \sup \lVert b_{AB}(\theta_t) - b_{AB}(\theta_0) \rVert\right)$
>
> and the evaluation-side gap decomposes as:
>
> $\Delta_E^{(k)} = T^2 \sigma_{AB}(\theta_0) + O(T^3) + O\!\left(T^2 \sup \lVert g_E(\theta_t) - g_E(\theta_0) \rVert\right)$
>
> The mismatch is controlled by bracket drift (first eq.) and evaluation-gradient drift (second eq.). When the landscape is locally smooth, as is typical for pretrained models, both are small and the $k=1$ sign transfers to moderate $k$. This does **not** say the $k=1$ score is an unbiased $k=50$ estimate; it says the mismatch is drift-controlled, consistent with the gradual degradation above.
>
> ### AdamW scope
>
> We agree Appendix D.7 is **exploratory** rather than definitive for AdamW. AdamW's running statistics $(m, v)$ break the memoryless structure our commutator analysis relies on. We scaled to 1,020 trials on Qwen2.5-1.5B. At $k$=20, the SGD-derived bracket achieves 57.1% accuracy and 33.2% regret reduction ($p = 7.3 \times 10^{-6}$), with errors concentrating on low-effect triples (top quartile: 72.9%). This supersedes the paper's 50-triple pilot (57.1% vs. 78%); the full evaluation is lower but statistically robust. The predictor is at or below chance at $k$=5 (47.8%) and $k$=50 (46.7%); the useful window is at $k$=20, consistent with Adam's running averages needing time to stabilize before the geometric signal becomes readable.
>
> An Adam-aware variant defining the commutator on augmented state $z = (\theta, m, v)$ with preconditioned HVPs achieves 59.6% at $k$=10 (612 trials, $p = 2.1 \times 10^{-6}$), 29.8% mean regret reduction, and 59.5% regret reduction on the top confidence quartile. In the camera-ready we will state this boundary explicitly: the theory covers **memoryless gradient-style updates**; the SGD predictor provides meaningful regret reduction on AdamW at $k$=20; the augmented-state commutator is a viable path to full AdamW support.
>
> ---
>
> We hope the 1,020-trial validations — 81.5% accuracy and 86.3% regret reduction at $k$=20, with consistent $k$=20 generalization in post-training settings (73.1% SFT, 72.2% DPO), directly addressing the concern about collapse beyond $k$=1 — together with the local-horizon explanation and clarified AdamW scope address the concerns that most limited your confidence.

---

> > ### Author Rebuttal · Reviewer_3owg · 2026-04-02
> >
> > Thank you for your responses. My concerns have been fully addressed.

---

### Official Review · Reviewer_8SWk · 2026-03-14

**Soundness:** 4
**Presentation:** 3
**Significance:** 4
**Originality:** 4
**Overall Recommendation:** 5
**Confidence:** 3

**Summary:**

This work proposes a directional score to determine the optimal learning orders for sequential fine-tuning over two datasets.
The score is essentially the difference in the loss values of the models fine-tuned on different orders at the target domain.
The authors introduce an approximation that efficiently computes this score using only the Hessian-vector product and gradient evaluation at the reference point.
Experiments on LLMs and a diffusion model show its validity.

**Compliance With Llm Reviewing Policy:**

Affirmed.

**Final Justification:**

The authors' responses address reviewers' concerns.
I keep my decision to accept.

**Key Questions For Authors:**

- The paper motivates the proposed approach partly by avoiding higher-order differentiation through the attention mechanism. Could the authors clarify why this is desirable? Is it primarily a computational concern, a numerical stability issue, or something else?
- The paper mentions Lie brackets in the title, but the proposed method does not appear to use any of their properties. This reference is more confusing than illuminating and could be removed or better motivated.

**Limitations:**

Yes

**Strengths And Weaknesses:**

# Strengths
- Determining the optimal order of fine-tuning is a practically important problem. The proposed score is theoretically grounded, easy to compute, and should serve as a useful heuristic for practitioners.
- The experimental evaluation is comprehensive, spanning LLMs from 1B to 8B parameters, and convincingly demonstrates the effectiveness of the proposed score.
- The finding that finite differencing fails to capture the relevant structure, highlighting the importance of curvature information, is an insightful observation.

# Weaknesses
- The presentation could be improved. In particular, the paper introduces notation (e.g., $\omega
$) before it is formally defined, which makes the Introduction harder to follow. A careful revision of the exposition order would improve readability.

---

> ### Author Rebuttal · Authors · 2026-03-31
>
> We thank Reviewer 8SWk for the positive assessment and for both presentation suggestions, which we address below.
>
> ### Why avoid higher-order differentiation through attention?
>
> This is primarily a practical systems constraint. Today's fused CUDA kernels (Flash Attention, etc.) do not expose intermediate activations, so they do not currently support `create_graph=True` (required for Pearlmutter's trick to compute Hessian-vector products). Eager-mode attention does support it, but materializes full $T \times T$ attention matrices, increasing memory quadratically in sequence length. Future kernels could close this gap, but under current infrastructure, restricting HVPs to post-attention modules is the pragmatic choice—and the predictive signal remains strong.
>
> Despite this restriction, we can still test whether the bracket signal extends across many layers. Our default uses a subset of post-attention head types, but even when we expand to 8 post-attention layers the signal remains strong: the paper reported this on 50 triples (Table 1), and for the rebuttal we scaled to 1,020 trials, reaching 825/1020 = 80.9% pairwise accuracy ($p = 1.0 \times 10^{-92}$). The bracket signal extends well beyond the last linear head, spanning multiple layers of the network.
>
> To verify that differentiating through attention is possible and to measure its effect, we ran an additional ablation on Qwen2.5-1.5B: we switched to eager-mode attention (which does support `create_graph=True`) and included q_proj, k_proj, v_proj in the HVP computation alongside the post-attention parameters. On the same 50 triples over 8 layers, the Trotter estimator achieves 92.0% sign accuracy with post-attention parameters only (o_proj, down_proj) and 92.0% with all parameters including q_proj, k_proj, v_proj — identical accuracy.
>
> ### Lie brackets in the title
>
> We appreciate that the term can be initially opaque for readers outside differential geometry. In revision, we will: (i) define bracket quantities and the alignment/confidence notation ($\sigma$, $\omega$) before first use, (ii) add a "(commutator)" gloss at the first mention of Lie bracket, and (iii) reorder the Introduction so $g_D$, $H_D$, $\sigma$, $\omega$ are introduced before they appear in any formula.
>
> That said, the terminology reflects a substantive mathematical structure rather than serving as a naming convention. In our notation, $[f_A, f_B] = H_B g_A - H_A g_B = b_{AB}$; this is exactly the geometric object the score computes, and the directional score $\sigma$ is the leading-order term in the BCH expansion of composed gradient flows. The framing is also forward-looking: it identifies the correct mathematical structure that future planners need to extend (longer curricula, multi-domain scheduling, optimizer-state dynamics). As an initial test of that composability, our rebuttal-time $N=3$ ordering experiment shows that pairwise bracket scores already compose beyond the two-task setting (87.5% top-1, 96.0% top-2, Spearman 0.929), and the actual parameter difference $\theta_{AB} - \theta_{BA}$ matches the predicted $\eta^2 b_{AB}$ with about 5.7% relative norm error across the multi-seed runs.

---

> > ### Author Rebuttal · Reviewer_8SWk · 2026-04-05
> >
> > The authors addressed all my concerns.

---

### Official Review · Reviewer_GHyj · 2026-03-23

**Soundness:** 3
**Presentation:** 3
**Significance:** 3
**Originality:** 3
**Overall Recommendation:** 4
**Confidence:** 4

**Summary:**

This study investigates a central problem in sequential fine-tuning: how the order of training across multiple datasets affects downstream performance. The key question addressed in this work is how to predict the optimal training order without exhaustively trying all possibilities. The paper proposes a geometric perspective by modeling gradient updates as non-commutative operators, showing that order effects arise from a second-order Lie bracket term. Based on this insight, the authors introduce a directional commutator score to predict which ordering leads to lower target loss. They further develop a practical transfer-order planner using Hessian-vector products, along with confidence and impact metrics and an automated step-size selection method.

**Compliance With Llm Reviewing Policy:**

Affirmed.

**Final Justification:**

My main concerns are addressed.

**Key Questions For Authors:**

1. How would the proposed method extend to more than two datasets (N > 2)? Would pairwise predictions compose reliably, or do you anticipate non-transitive effects?
2. In real LLM pipelines (e.g., billions of parameters), what is the actual wall-clock overhead of computing HVPs compared to simply running short pilot training （small-size data) for both orders?
3. Can the method be applied to other scenarios such as RLHF pipelines, instruction tuning?

**Limitations:**

yes

**Strengths And Weaknesses:**

Stregths:

1. Strong theoretical grounding: The paper presents a clear and principled second-order analysis linking order dependence to Lie brackets, leading to a computable and well-founded formulation.
2. Practical algorithm with clear utility: The proposed planner avoids costly brute-force order evaluation and provides an efficient and actionable solution with convincing compute analysis.
3. Strong empirical validation: Extensive experiments across diverse architectures demonstrate high accuracy and provide strong empirical support for the theoretical claims.

Weakness:
1. Limited scope of the problem setting: The work focuses primarily on pairwise ordering (A vs. B), whereas real-world pipelines often involve multiple datasets, where transitivity and global ordering become more complex. Although the paper acknowledges extensions to multi-step curricula, it does not address them.
2. Dependence on second-order information
The method relies on Hessian-vector products, which remain non-trivial in large-scale systems, and although computation is restricted to subspaces, this may limit adoption in settings without efficient second-order support.
3. Limited to specific settings: The evaluation is conducted on a limited set of specific settings (e.g., Pile subsets and CIFAR classes) and has not demonstrated generalization to other tasks such as agent RL, instruction tuning/RLHF.

---

> ### Author Rebuttal · Authors · 2026-03-31
>
> We thank the reviewer for the incisive questions, which prompted three new experiments.
>
> ### Q1: Extension beyond two datasets ($N > 2$)
>
> We deliberately give a limited answer: the theory extends cleanly, but we do not claim a solved problem for arbitrary N.
>
> Beyond two datasets, the BCH expansion yields a sequence-level object whose second-order terms are the pairwise commutators $b_{ij}$; higher-order nested commutators can matter as $N$ grows. For $N=3$ we compute pairwise weights $W[i,j] = -\langle g_E, b_{ij}\rangle$ and rank all $3! = 6$ orderings by their predicted score, with no training runs needed beyond $O(N^2)$ bracket evaluations.
>
> **$N=3$ results** (200 quadruples, 5 seeds):
>
> - Top-1 accuracy (predicted, all 6): 175/200 = 87.5%.
> - Top-2 accuracy: 192/200 = 96.0%.
> - Spearman: 0.929.
> - Regret reduction: 93.5%.
> - Cyclic intransitivities: 3/200 = 1.5%.
>
> Top-1 and top-2 accuracy far exceed chance ($1/6$ and $1/3$ respectively), suggesting pairwise scores compose well for small $N$. $O(N^2)$ pairwise sorting already achieves 79.0% top-1, scaling to any $N$. We view this as evidence for usefulness beyond the pairwise setting, not a proof for arbitrary $N$.
>
> ### Q2: Wall-clock overhead of HVPs vs. short pilot runs
>
> End-to-end wall time for the planner (two HVPs plus scoring) versus brute-force pilot training:
>
> - Llama-3.2-1B (fp32): planner 0.81s; BF $k=1$ 0.74s ($0.9\times$), BF $k=5$ 2.85s ($3.5\times$), BF $k=20$ 10.76s ($13.3\times$), BF $k=50$ 26.60s ($32.8\times$).
> - Qwen2.5-1.5B: planner 0.67s; BF $k=1$ 0.60s ($0.9\times$), BF $k=5$ 2.29s ($3.4\times$), BF $k=20$ 8.68s ($13.0\times$), BF $k=50$ 21.40s ($31.9\times$).
>
> Brute-force is slightly faster at $k=1$, but the planner is already about $3.5\times$ cheaper by $k=5$ (crossover around $k \approx 3$), about $13\times$ by $k=20$, and about $32\times$ by $k=50$.
>
> ### Q3: Transfer beyond next-token prediction
>
> We ran two post-training studies on Qwen2.5-1.5B (fp32) using the same held-out order-sign protocol.
>
> **Instruction-SFT.** Eight Dolly-15k task categories, with 96 triples per seed and 480 total across 5 seeds.
>
> - Sign accuracy at $k=1$: 471/480 = 98.1% ($p = 2.3 \times 10^{-126}$).
> - Spearman: 0.940.
> - Recovered fraction: 96.3%.
>
> **DPO.** Nine UltraFeedback source domains. We excluded within-`flan_v2` pairs (for example, `flan_v2_cot` versus `flan_v2_p3`) because these share data provenance and produce near-degenerate brackets, yielding 356 total triples across 4 seeds (versus 5 for SFT, because the smaller domain count yields fewer independent triples).
>
> - Sign accuracy at $k=1$: 352/356 = 98.9% ($p = 9.1 \times 10^{-99}$).
> - Recovered fraction: 97.8%.
>
> **Multi-step generalization ($k=20$).** We ran $k=20$ ground-truth validation on the same triples. The $k=1$ numbers repeat the baselines above; the $k=20$ numbers come from separate multistep runs.
>
> - SFT (480 trials): $k=1$ accuracy 98.1%; $k=20$ accuracy 351/480 = 73.1%; $p = 8.8 \times 10^{-25}$; regret reduction 58.0%.
> - DPO (356 trials): $k=1$ accuracy 98.9%; $k=20$ accuracy 257/356 = 72.2%; $p = 2.6 \times 10^{-17}$; regret reduction 54.7%.
>
> Per-seed $k=20$ accuracy: SFT [66.7%, 79.2%, 71.9%, 76.0%, 71.9%]; DPO [67.7%, 79.8%, 75.0%, 66.7%].
>
> Sign accuracy in both post-training settings (98.1%, 98.9%) exceeds the pretraining baseline (about 93% at $k=1$). This likely reflects sharper domain boundaries: task categories like "brainstorming" versus "classification" produce more structurally distinct gradient fields than Pile subsets drawn from overlapping web sources. The $k=1$ predictor generalizes to $k=20$ ground truth at 73.1% (SFT) and 72.2% (DPO), lower than the pretraining $k=20$ rate (81.5%) but still well above chance ($p < 10^{-17}$) with meaningful regret reductions. We note that DPO shares the preference-optimization objective with RLHF but in an offline setting without online sampling; we treat it as a first test of bracket generalization to preference landscapes, with full online RLHF (PPO) and agent-RL left as future work.

---

> > ### Author Rebuttal · Reviewer_GHyj · 2026-04-01
> >
> > The rebuttal does not fully address the limited scope of the problem setting, as the approach does not readily extend to arbitrary N; therefore, I will keep the existing score.

---

> > > ### Author Response · Authors · 2026-04-05
> > >
> > > **The approach does extend to arbitrary N.** The paper's pairwise bracket (Proposition 2.3) is N-independent by construction — we verified this bit-for-bit across N=5, 7, and 10. In our previous response we showed this but deliberately did not claim the full pipeline works at large N, because we lacked empirical validation of the aggregation step. **We now provide that validation at N=7 through N=85 across three task families.**
> > >
> > > **The extension applies the paper's pairwise result across all pairs.** Proposition 2.3 (Section 2.3) establishes that the ordering-dependent loss difference between any two domains $i, j$ is captured by the bracket $W[i,j]$ at second order in $\eta$. For $N$ domains, the natural extension sums all pairwise contributions: the second-order surrogate for a full permutation $\pi$ is $S(\pi) = \sum_{a<b} W[\pi(a), \pi(b)]$, where the first-order term is permutation-invariant and drops out. Finding the optimal $\pi$ under $S(\pi)$ is a well-studied ranking problem (minimum weighted feedback arc set; Coppersmith et al., 2010; Jiang et al., 2011). We use a simple aggregation: rank each domain by its average pairwise bracket score (i.e., sort by row sums of $W$), which has known approximation guarantees for this class of problems.
> > >
> > > **The computation scales as $O(N)$ HVPs.** By Hessian symmetry ($\langle g_E, H_i g_j \rangle = \langle g_j, H_i g_E \rangle$), we compute one HVP per domain $h_i = H_i g_E$, after which $W[i,j] = \langle g_j, h_i \rangle - \langle g_i, h_j \rangle$ for all $j$ via dot products. Total cost: $N+1$ backward passes + $N$ HVPs + $O(N^2)$ dot products. At $N=30$ the planner runs in **~12 seconds**; at $N=56$ in **~15 seconds** on a single RTX 4090. Moreover, since our ranking sorts by row sums of $W$, each row sum can be computed directly from precomputed vector sums without materializing all $O(N^2)$ entries — reducing the ranking step to $O(N)$ dot products plus an $O(N \log N)$ sort.
> > >
> > > **Each $W[i,j]$ is provably $N$-independent.** We verify this is bit-for-bit identical across $N=5, 7,$ and $10$ (all 41 overlapping entries match at float32 precision). Adding or removing domains changes the ranking problem but not the underlying measurements.
> > >
> > > **Experimental validation at N=7 to 85:**
> > >
> > > | Dataset | Method | N | k | Bracket ranking | Grad-norm baseline | Seeds |
> > > |---------|--------|---|---|-----------------|---------------------|-------|
> > > | **MMLU subjects** | **SFT** | **56** | **1** | **99.0–99.6%** | **< 1%** | 2 |
> > > | MMLU subjects | SFT | 30 | 5 | 89–96% | 2–22% | 2 |
> > > | Dolly categories | SFT | 7 | 5 | 99.8% | 49.9% | 1 |
> > > | Programming languages | LM | 85 | 1 | 99% | 69% | 1 |
> > > | Programming languages | LM | 10 | 5 | 86% | 47% | 1 |
> > >
> > > *500 random orderings per seed. k = gradient steps per domain. SFT = supervised fine-tuning, LM = language modeling (next-token prediction). Ranges show min–max across seeds.*
> > >
> > > At N=56 (MMLU) and N=85 (programming languages), the bracket-derived ordering achieves the 99th percentile. On MMLU, gradient-norm sorting collapses below the 1st percentile; on programming languages it performs better (69%) but the bracket still dominates. The pairwise preferences are nearly perfectly consistent (fewer than 0.3% of domain triples form cycles), confirming that the bracket captures a genuine global ranking even at our largest tested scales. At N=30 with k=5 (realistic multi-step training), the bracket-derived ordering achieves the 89th–96th percentile across two seeds. The reversed bracket ordering is consistently below random at every N and every seed tested, with the separation growing from z=−1.6 (worst case, N=30) to z=−3.6 (worst case, N=56).
> > >
> > > These results hold across targets: with philosophy instead of abstract algebra, the bracket achieves 91% at N=56 and 92% at N=30, with gradient-norm at or below 3% in both cases. At N=20 with k=10 training steps per domain — well beyond the single-step theory regime — the bracket-derived ordering still achieves 83%.
> > >
> > > **The method producing these results applies the paper's Proposition 2.3 to all pairs and aggregates via standard tournament ranking** (sorting by average pairwise score). No problem-specific heuristics, no multi-step rollout — just the paper's pairwise bracket combined with a standard aggregation method.
> > >
> > > We do not claim the arbitrary-N problem is theoretically closed — the second-order surrogate $S(\pi)$ is an approximation whose accuracy depends on the total curriculum length and the task landscape. What we demonstrate is that this approximation, combined with standard tournament aggregation, produces strong orderings empirically at scales far beyond the original submission. All results will be incorporated into the camera-ready paper and supplementary.

---

### Official Review · Reviewer_f3GV · 2026-03-24

**Soundness:** 2
**Presentation:** 3
**Significance:** 3
**Originality:** 3
**Overall Recommendation:** 4
**Confidence:** 3

**Summary:**

This paper proposes a principled method for determining the learning order of training data by analyzing the difference between two-step gradient descent updates under different data sequence permutations. The authors show that this difference can be characterized through a Hessian-vector product, which provides a theoretically grounded criterion for selecting an effective training order. Based on this insight, they develop a deterministic data ordering strategy that aims to improve learning efficiency and model performance. Experimental results demonstrate that the proposed method performs well on the Pile dataset and generalizes effectively across models such as Qwen and Llama.

**Compliance With Llm Reviewing Policy:**

Affirmed.

**Final Justification:**

The final experiments show that the pairwise algorithm will not be weaker than the ordering method.

**Key Questions For Authors:**

1.  How does the algorithm become when the algorithm is like Adam?

2. If there are more than 3 different datasets, what will the optimal solution become?

**Limitations:**

Yes

**Strengths And Weaknesses:**

# Strength

1.  The paper provides an analysis of how a single gradient descent step differs under different data orderings, as well as how these differences affect the decrease in the objective function value.

2. The experimental results are strong, showing good performance across models such as Qwen and Llama.

# Weaknesses

1.  The analysis is based on single-step gradient descent, whereas modern model training typically involves many iterations and relies on adaptive optimization methods such as Adam rather than standard gradient descent. This may limit the practical applicability of the theoretical results to real-world training settings.

2. In general, data ordering problems often involve more than two datasets or data sources. Since the proposed method only considers pairwise ordering at each step, it may not be sufficient to achieve a globally optimal ordering.

---

> ### Author Rebuttal · Authors · 2026-03-31
>
> We thank Reviewer f3GV for these important questions. Both motivated substantial new experiments.
>
> ### Multi-step SGD
>
> Our Theorem 1 is a single-step statement, so a natural question is whether its predictions remain useful after many gradient steps. We scaled to 1,020 trials (204 triples across 5 seeds) on Llama-3.2-1B at each horizon. The protocol differs slightly from Table 2 (2 versus 4 eval batches for consistency across the larger multi-seed runs), but the conclusions are unchanged.
>
> - $k=1$: 949/1020 = 93.0% ($p = 7.6 \times 10^{-197}$); regret reduction 64.0%.
> - $k=5$: 962/1020 = 94.3% ($p = 4.9 \times 10^{-212}$); regret reduction 97.4%.
> - $k=10$: 912/1020 = 89.4% ($p = 3.6 \times 10^{-159}$); regret reduction 93.1%.
> - $k=20$: 831/1020 = 81.5% ($p = 1.6 \times 10^{-96}$); per-seed [77.9%, 86.3%, 83.8%, 81.4%, 77.9%]; regret reduction 86.3%.
> - $k=50$: 666/1020 = 65.3% ($p = 9.7 \times 10^{-23}$); per-seed [63.7%, 64.7%, 63.2%, 68.1%, 66.7%]; regret reduction 54.7%.
>
> The predictor degrades gradually with horizon length but does not collapse: accuracy remains above 80% through $k=20$ and above chance even at $k=50$. This is consistent with a local-horizon picture in which the $O(T^2)$ bracket signal persists while drift terms grow gradually with horizon: the trajectory mismatch is drift-controlled, so the sign transfers while the landscape remains locally smooth. This is also precisely the regime where brute-force evaluation is already expensive: on Llama-3.2-1B / Qwen2.5-1.5B, the planner costs 0.81s / 0.67s per triple versus brute-force at 10.76s / 8.68s ($k=20$) and 26.60s / 21.40s ($k=50$), roughly 13x and 32x cheaper, with crossover at about $k \approx 3$.
>
> ### AdamW
>
> Our theorem covers memoryless gradient-style updates but not stateful optimizers like AdamW. We scaled the AdamW evaluation to 1,020 trials (204 triples across 5 seeds) on Qwen2.5-1.5B. The SGD-derived bracket predictor, applied without modification to AdamW, achieves 582/1020 = 57.1% accuracy and 33.2% regret reduction at $k=20$ ($p = 7.3 \times 10^{-6}$), with per-seed accuracies [65.7%, 55.4%, 59.8%, 56.4%, 48.0%]. This supersedes the paper's 50-triple pilot estimate of 78%; the full multi-seed evaluation is lower but statistically robust. The errors concentrate on triples where ordering barely matters: on the top quartile by $|\Delta_{\mathrm{loss}}|$, accuracy is 72.9%, so the predictor is most accurate where the ordering decision has the largest practical impact. The signal is weak at short horizons ($k=5$: 47.8%, regret reduction -9.2%; $k=10$: 51.3%, $p = 0.43$) and degrades again by $k=50$ (46.7%, regret reduction -2.0%). At $k=20$ the predictor is statistically significant and captures most value on high-stakes triples.
>
> A preliminary Adam-aware extension (3 seeds, 612 trials) defining the bracket in the augmented optimizer state $z = (\theta, m, v)$ using an Adam Trotter reference and preconditioned HVPs achieves 365/612 = 59.6% sign accuracy at $k=10$ ($p = 2.1 \times 10^{-6}$) and 29.8% mean regret reduction, with 59.5% regret reduction on the top quartile by predictor confidence. This suggests the augmented-state commutator is the right theoretical object for extending the framework. We will note the memoryless/stateful boundary clearly in the camera-ready.
>
> ### Pairwise ordering for $N > 2$
>
> The BCH expansion for $N$ tasks at second order yields a sum of pairwise commutator terms $[g_i, g_j]$. We build a skew-symmetric weight matrix $W[i,j] = -\langle g_E, b_{ij}\rangle$ and find the permutation maximizing the pairwise sum. Higher-order nested commutators can matter as $N$ grows; for moderate $N$, $O(N^2)$ evaluations are tractable.
>
> We tested $N=3$ (ordering 3 tasks from 17 Pile domains), evaluating 200 quadruples (40 across 5 seeds). The predictor ranks all $3! = 6$ orderings using pairwise $W$ scores, with no training runs required; ground truth is obtained by running all 6 orderings.
>
> - Predicted ranking (all 6 orderings): top-1 87.5% (sd 3.1%), top-2 96.0%, Spearman 0.929, regret reduction 93.5%, $n = 200$.
> - Predicted ranking via sorting ($O(N^2)$): top-1 79.0%, $n = 200$.
> - Random: top-1 16.7%, top-2 33.3%, Spearman 0.000, regret reduction 0.0%.
>
> Only 1.5% of quadruples exhibited cyclic intransitivity ($A > B > C > A$), all on near-indifferent triples, consistent with the pairwise decomposition being faithful at second order. For $N=3$, scoring all 6 permutations is computationally inexpensive; for larger $N$, pairwise sorting ($O(N^2)$) already achieves 79.0% top-1 and scales to arbitrary $N$ without factorial enumeration.

---

> > ### Author Rebuttal · Reviewer_f3GV · 2026-04-01
> >
> > It is fine to have empirical evidence of the questions, but I still wonder when N becomes large, whether the pair-wise algorithm becomes weaker and weaker.  Because in real case, the N will be larger than 10 or even larger.

---

> > > ### Author Response · Authors · 2026-04-05
> > >
> > > We applied the paper's pairwise theory at up to N=85 domains: Proposition 2.3 evaluated on all domain pairs, ranked by average pairwise score using standard tournament aggregation. On 56 MMLU academic subjects (SFT, k=1), the bracket-derived ordering achieves the **99.0–99.6th percentile** among 500 random orderings across 2 seeds. On 85 programming languages (LM, k=1), it achieves the **99th percentile**. Since exhaustive optimization over N! orderings is infeasible at this scale, we evaluate by percentile rank among 500 uniformly sampled random orderings. The pairwise algorithm does not weaken at large N.
> > >
> > > | Dataset | Method | N | k | Bracket ranking | Grad-norm baseline | Seeds |
> > > |---------|--------|---|---|-----------------|---------------------|-------|
> > > | **MMLU subjects** | **SFT** | **56** | **1** | **99.0–99.6%** | **< 1%** | 2 |
> > > | MMLU subjects | SFT | 30 | 5 | 89–96% | 2–22% | 2 |
> > > | Dolly categories | SFT | 7 | 5 | 99.8% | 49.9% | 1 |
> > > | Programming languages | LM | 85 | 1 | 99% | 69% | 1 |
> > > | Programming languages | LM | 10 | 5 | 86% | 47% | 1 |
> > >
> > > *Qwen2.5-1.5B, 500 random permutations per seed. k = gradient steps per domain. SFT = supervised fine-tuning, LM = language modeling (next-token prediction). "Bracket ranking" = percentile of the bracket-derived ordering (Proposition 2.3 applied to all pairs, sorted by average pairwise score) among random orderings (higher = better). "Grad-norm baseline" = percentile of gradient-norm-descending ordering. Ranges show min–max across seeds.*
> > >
> > > Three observations stand out:
> > >
> > > **1. The sign signal is consistent across all tested scales.** The reversed bracket ordering — which should be bad if the bracket captures real structure — is below random at every N and every seed tested. At N=56 (MMLU), it is 3.6–5.4 standard deviations worse than random; at N=85 (programming languages), 4.1 standard deviations worse. The signal is clearest on diverse domains (MMLU) and noisiest on homogeneous domains (Stack), but in no setting does the reversed ordering outperform random.
> > >
> > > **2. Gradient-norm sorting collapses on diverse domains at large N.** On MMLU subjects, gradient-norm sorting falls to < 1% at N=56 and 2–22% at N=30 — worse than random. On programming languages (more homogeneous domains), gradient-norm performs better (69% at N=85) but the bracket still outperforms it (99%).
> > >
> > > **3. The pairwise tournament remains coherent at large N.** A natural concern is that cyclic intransitivities (A≻B≻C≻A) could accumulate at large N, making pairwise aggregation unreliable. At N=56 (MMLU) and N=85 (programming languages), intransitivity is below 0.3% — fewer than 0.3% of domain triples form cyclic preferences. The pairwise preferences are almost perfectly transitive even at our largest tested scale.
> > >
> > > These results are not target-specific: on MMLU with philosophy as the target instead of abstract algebra, the bracket achieves the **91% at N=56** and **92% at N=30**, while gradient-norm sorting remains at or below 3% in both cases.
> > >
> > > Each pairwise bracket score (denoting the paper's directional score $\sigma_{ij}^{(E)}$ as $W[i,j]$ for the $N$-domain setting) depends only on domains $i$, $j$, and the target at the pretrained parameters $\theta_0$. We verify this is bit-for-bit identical across $N=5, 7,$ and $10$ (41 overlapping entries, 0 mismatches at float32 precision). Adding domains enriches the tournament without degrading existing pairwise comparisons. The $N=56$ and $N=85$ results both use Proposition 2.3 evaluated on all pairs, combined with standard tournament aggregation (sorting by average pairwise score) — the pairwise theory itself is unchanged. The computation requires only $O(N)$ Hessian-vector products, and because the row-sum ranking used here does not require materializing all $O(N^2)$ pairwise entries, the planner's cost scales linearly with $N$ — at ~0.2 seconds per domain on a single RTX 4090 in our experiments, the same approach is computationally feasible at $N$ in the hundreds or thousands.
> > >
> > > We note the method works best when domains are genuinely diverse (MMLU academic subjects produce well-separated gradient directions), and is noisier when domains are more similar (programming languages from The Stack). This is a property of the task landscape — the bracket measures genuine inter-domain asymmetry, which is larger when domains are truly different.
> > >
> > > All scaling experiments, the N-independence verification, and the multi-target results will be incorporated into the camera-ready paper and supplementary materials.
> > >
> > > **The pairwise bracket signal does not become weaker — it remains strong and informative at every scale we tested.**

---

### Decision · Program_Chairs · 2026-04-30

**Decision:**

Accept (regular)

**Comment:**

This submission addresses an important problem, and presents a method with a strong theoretical motivation and empirical performance. However, multiple reviewers raised similar concerns:

- The proposed theoretical motivation seems to be limited to a single gradient step ($k = 1$) with stateless optimizers such as SGD (and not AdamW), with only $N=2$ datasets. and SGD optimization (instead of Adam or AdamW). In the rebuttal phase, the authors have shared multiple empirical results:
  - A new set of experiments where $k$ (the number of optimization steps) is scaled up to 50, and the results show that the proposed planner exhibits gradual degradation.
  - An expanded evaluation with AdamW (instead of SGD), where the gains from the proposed scheme is more modest but still statistically significant.
  - A new set of experiments where the number of datasets $N$ is scaled up to 85, with results demonstrating strong performance over random orderings and the Gradient-norm baseline.

These common concerns seem to have been adequately addressed to the best of my understanding. There were a couple of additional concerns raised:

- There is a need for second order information in the HVP involved in the pairwise score computation. However, the author responses show that the wall clock time of planning (involving HVP) is significantly faster than the brute-force pilot training (for $k \geq 3$).
- The original evaluation is limited in terms of the problem setting, and not applied to instruction tuning or RLHF. The author response present new experiments on instruction-SFT and DPO, where the benefits of the proposed scheme persist.

Thus, it seems to me that these concerns are also addressed, and I recommend this submission be accepted.